# Human LINE-1 retrotransposition requires a metastable coiled coil and a positively charged N-terminus in L1ORF1p

**Elena Khazina, Oliver Weichenrieder\***

Department of Biochemistry, Max Planck Institute for Developmental Biology, Tübingen, Germany

**Abstract** LINE-1 (L1) is an autonomous retrotransposon, which acted throughout mammalian evolution and keeps contributing to human genotypic diversity, genetic disease and cancer. L1 encodes two essential proteins: L1ORF1p, a unique RNA-binding protein, and L1ORF2p, an endonuclease and reverse transcriptase. L1ORF1p contains an essential, but rapidly evolving N-terminal portion, homo-trimerizes via a coiled coil and packages L1RNA into large assemblies. Here, we determined crystal structures of the entire coiled coil domain of human L1ORF1p. We show that retrotransposition requires a non-ideal and metastable coiled coil structure, and a strongly basic L1ORF1p amino terminus. Human L1ORF1p therefore emerges as a highly calibrated molecular machine, sensitive to mutation but functional in different hosts. Our analysis rationalizes the locally rapid L1ORF1p sequence evolution and reveals striking mechanistic parallels to coiled coil-containing membrane fusion proteins. It also suggests how trimeric L1ORF1p could form larger meshworks and indicates critical novel steps in L1 retrotransposition.

DOI: https://doi.org/10.7554/eLife.34960.001

**\*For correspondence:**
oliver.weichenrieder@tuebingen.mpg.de

## Introduction

The mammalian LINE-1 (long interspersed element 1, L1) retrotransposon has had a considerable impact on the evolution of mammalian genome organization and continues to shape the evolution of the human genome. Roughly 17% of the human genome sequence corresponds to fragments or full-length L1 copies of different evolutionary age, contrasting with only about 1.5% of our genome, which encodes all of the human proteins (*Lander et al., 2001*; *Stewart et al., 2011*). L1 is the only autonomously active mobile genetic element in the human genome, but also mobilizes non-autonomous Alu and SVA elements (*Garcia-Perez et al., 2016*; *Goodier, 2016*; *Mita and Boeke, 2016*; *Richardson et al., 2015*). Autonomous retrotransposition relies on two L1-encoded proteins. The L1ORF1 protein (L1ORF1p) is known as an RNA-binding protein (*Hohjoh and Singer, 1996*; *Martin, 1991*), whereas the L1ORF2 protein (L1ORF2p) harbors the necessary catalytic functions, consisting of an endonuclease and a reverse transcriptase (*Feng et al., 1996*; *Kazazian et al., 1988*; *Moran et al., 1996*).

L1 propagates via an RNA intermediate in a 'copy-and-paste' fashion. It does not rely on long terminal repeats (LTRs) for the reverse transcription and genome integration steps, in contrast to LTR retrotransposons and retroviruses (*Sultana et al., 2017*). Hence classified as a non-LTR retrotransposon, L1 integrates via target-primed reverse transcription, a telomerase-like mechanism, where the reverse transcription of L1RNA occurs directly at the spot of genomic integration (*Cost et al., 2002*; *Luan et al., 1993*). It is poorly understood, however, how L1RNA, as a part of large L1 ribonucleoprotein particles (L1RNPs), gains access to the chromatin in dividing (*Mita et al., 2018*) as well as non-dividing cells (*Kubo et al., 2006*; *Macia et al., 2017*).

**eLife digest** Almost half of the human genome consists of DNA strings that have been copied and pasted from one part of the genome to another many thousands of times. These strings of DNA are called mobile genetic elements. Mobile elements can disrupt important genes, causing disease and cancer, but they can also drive evolution.

Presently, only one type of mobile element, called LINE-1, is active in the human genome and able to multiply without help from other mobile elements. LINE-1 DNA is 'transcribed' to form molecules of LINE-1 RNA, which can then be 'translated' into two distinct proteins. These bind to LINE-1 RNA, which then gets back-transcribed into DNA and inserted as a new LINE-1 element in a new region of the genome. One of the two proteins, called L1ORF1p, forms complexes where three copies of the protein come together. These 'trimers' cover and protect LINE-1 RNA and are required for LINE-1 mobility.

Different versions of L1ORF1p are found in different animals. Part of the protein is the same across all mammals, and this 'conserved' part controls the ability of L1ORF1p to bind to RNA. The non-conserved part of L1ORF1p differs even between humans and their closest animal relatives and little was known about its structure or role. However, this rapidly evolving part of L1ORF1p is essential for LINE-1 mobility.

Using X-ray crystallography, Khazina and Weichenrieder obtained a molecular snapshot of the part of L1ORF1p that interacts with other copies of the protein to form trimers. Combined with earlier snapshots of L1ORF1p's conserved part, this generated a complete structural model of the L1ORF1p trimer. Additional biophysical characterizations suggest that L1ORF1p trimers form a semi-stable structure that can partially open up, indicating how trimers could form larger assemblies of L1ORF1p on LINE-1 RNA. Indeed, the need to maintain a semi-stable structure could explain why L1ORF1p is evolving so rapidly. A second important finding is that the beginning of L1ORF1p needs to be positively charged – a requirement that warrants further exploration.

The structural and mechanistic insight into L1ORF1p points to critical new steps in LINE-1 mobilization. It will help to design inhibitor molecules with the goal to halt the mobilization process at various points and to dissect such steps in great detail. Understanding how to control LINE-1 mobility could help to improve stem cell therapies and reproduction assistance techniques, due to the fact that LINE-1 mobility is a potential source of mutation in stem cells, egg and sperm cells, and newly formed embryos.

DOI: https://doi.org/10.7554/eLife.34960.002

Retrotransposition must occur in germline cells in order to assure a lineage-specific, vertical transmission of L1 and its long-term survival in mammalian genomes. L1RNA and L1ORF1p are expressed in both gametogenesis and the early embryo (*Branciforte and Martin, 1994*; *Malki et al., 2014*; *Packer et al., 1993*; *Trelogan and Martin, 1995*), where early embryonic integrations lead to mosaic offspring (*Kano et al., 2009*; *van den Hurk et al., 2007*). Furthermore, retrotransposition also happens in somatic cells, such as in neuronal progenitor cells (*Coufal et al., 2009*; *Faulkner and Garcia-Perez, 2017*; *Muotri et al., 2005*). As a consequence of both germline and somatic insertions, L1 activity contributes to inter-individual human variation and diversity, but also causes genetic disease and cancer (*Burns, 2017*; *Hancks and Kazazian, 2016*; *Scott and Devine, 2017*). Importantly, human L1 expression and retrotransposition appears to be triggered in certain cancer types (*Carreira et al., 2014*; *Scott and Devine, 2017*) as well as in induced pluripotent stem cells (*Klawitter et al., 2016*; *Wissing et al., 2012*), as detected by the expression of L1ORF1p (*Klawitter et al., 2016*; *Rodić et al., 2014*; *Wissing et al., 2012*). Hence, considering the possible implications of L1 retrotransposition for human health and for the applications of stem cells in medicine and research, it is surprising how little we know about the mechanistic details of L1 retrotransposition.

Intriguingly, not only does L1 retrotransposition depend on the catalytic activity of the L1ORF2p (*Feng et al., 1996*; *Moran et al., 1996*), but also on an intact open reading frame encoding L1ORF1p (*Moran et al., 1996*). Multiple copies of L1ORF1p associate 'in cis' (*Basame et al., 2006*; *Kulpa and Moran, 2005*; *Sokolowski et al., 2017*; *Taylor et al., 2013*; *Wei et al., 2001*) with their

encoding L1RNA molecule, and the resulting L1RNP is considered as a functional intermediate in the retrotransposition process (*Hohjoh and Singer, 1996*; *Kulpa and Moran, 2005*; *Martin, 1991*; *Taylor et al., 2013*). Furthermore, L1ORF1p was shown to facilitate the rearrangement of nucleic acid structure and hence might be important as a 'nucleic acid chaperone' for remodeling the L1RNP (*Martin and Bushman, 2001*). Indeed, most of the published experimental data characterizes functions of L1ORF1p that are related to its interaction with RNA, whereas little is known about other roles of this protein in L1 retrotransposition.

Mammalian L1ORF1p has a unique architecture, even among the ORF1ps encoded by non-LTR retrotransposons (*Kapitonov and Jurka, 2003*; *Khazina and Weichenrieder, 2009*; *Schneider et al., 2013*). It consists of three structural domains, connected by short linkers. These domains are first, a coiled coil domain, which causes the protein to form homotrimers (*Martin et al., 2003*), second, an RRM (RNA recognition motif) domain, and third, a C-terminal domain (CTD), which cooperates with the RRM domain in binding single stranded nucleic acid substrates (*Januszyk et al., 2007*; *Khazina et al., 2011*; *Khazina and Weichenrieder, 2009*). In the case of the human protein, the coiled coil domain is preceded by a 51 residue long N-terminal region (NTR), harboring two serine-proline motifs that are known phosphorylation sites (*Cook et al., 2015*). Finally, there is also a short 15 residue tail at the C-terminal end of L1ORF1p that can be partially truncated without functional consequences (*Alisch et al., 2006*).

A series of crystal structures has uncovered the three-dimensional arrangement of the individual domains in the context of the L1ORF1p trimer (*Khazina et al., 2011*). The structures were obtained from an N-terminally truncated protein, lacking the NTR and the N-terminal half of the coiled coil domain, but they revealed L1ORF1p to be a highly structured and remarkably flexible RNA-binding protein. It became clear how the coiled coil domain mediates the trimerization of the protein and how it allows for the flexible attachment and organization of the RRM and CTD domains, such that between 27 and 45 nucleotides of single stranded RNA are bound and covered by one trimer (*Khazina et al., 2011*). However, although the structures rationalized trimerization and RNA binding of L1ORF1p, the N-terminally truncated protein was not able to promote L1 retrotransposition when tested in HeLa cells (*Khazina et al., 2011*).

We therefore decided to investigate the structural and mechanistic properties of the poorly conserved N-terminal sequences in L1ORF1p, and to which degree they contribute to L1 retrotransposition. To this aim, we combined biophysical with cell-based techniques and determined crystal structures for the entire coiled coil domain of the human L1ORF1p, enabling us to construct a composite model for the complete trimer. Surprisingly, in order to function in retrotransposition, the coiled coil apparently needs to be able to switch between fully structured and partially unstructured states. This requirement for metastability can explain the presence and delicate balance of both stabilizing and destabilizing elements in the structure of the coiled coil and the strong sensitivity to mutation. Finally, we also identified the positively charged amino terminus of L1ORF1p as an independent and novel determinant for L1 retrotransposition, a feature that is preserved in the mammalian homologs.

Consequently, L1ORF1p emerges as a delicate but remarkably autonomous protein regarding its host cell molecular environment, and with functions that clearly extend beyond RNA packaging. It shows striking parallels to other dynamic coiled coil proteins, which act in membrane fusogenic processes (*Skehel and Wiley, 1998*), hinting at presently uncharacterized steps in the L1 retrotransposition cycle.

## Results

### The N-terminal regions and coiled coil domains of mammalian L1ORF1 proteins show high sequence variability

Non-LTR retrotransposons encode ORF1 proteins with highly diverse architectures and distinct structural domains, frequently suggestive of RNA binding but possibly also of lipid or membrane interaction (*Khazina and Weichenrieder, 2009*; *Schneider et al., 2013*). The most commonly shared feature is, however, the apparent presence of coiled coil forming regions, suggesting self-association and oligomerization into dimeric, trimeric or higher order assemblies (*Figure 1A*, *Figure 1—figure supplement 1A*).

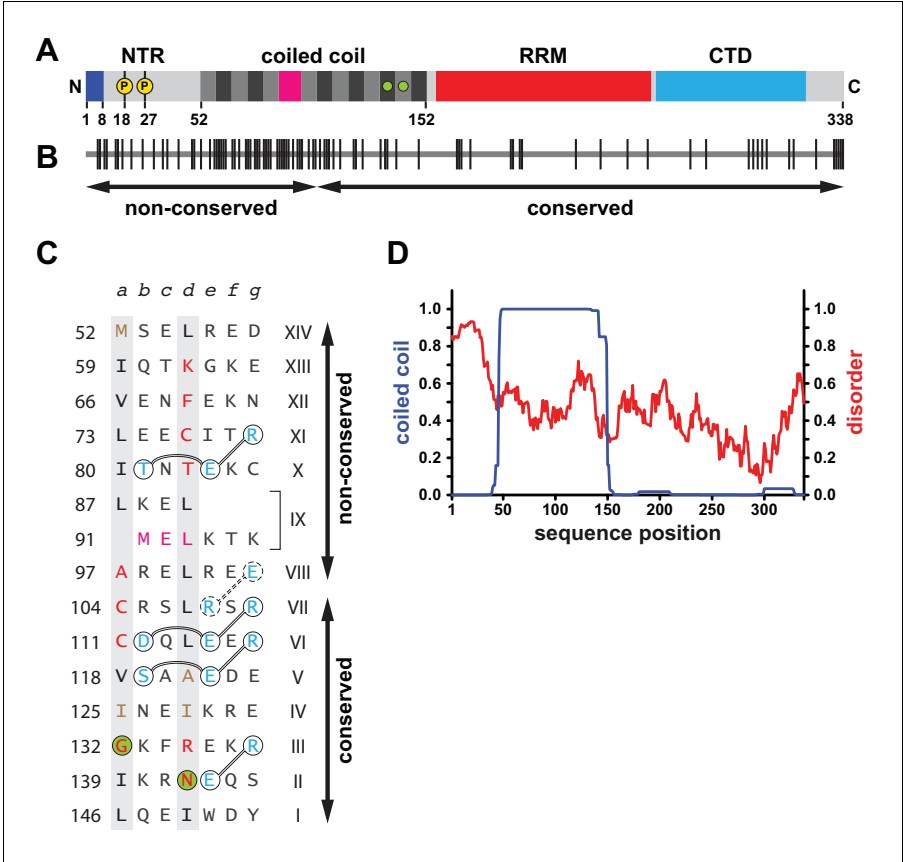

**Figure 1.** Sequence properties of human L1ORF1p. (**A**) Domain organization of human L1ORF1p, including findings from this work. See also *Figure 1—figure supplement 1*. The bar diagram shows the NTR in light grey with phosphorylation sites marked in yellow (*Cook et al., 2015*) and with the positively charged N-terminal residues in blue (this work). The coiled coil domain shows heptads alternating grey and black. The irregular heptad IX harbors a stammer and is in magenta. Ion-coordinating heptads are marked by green circles. The RRM domain is in red and the CTD is in cyan. (**B**) Conservation and evolution of human L1ORF1p. Non-conserved amino acid positions from an alignment of primate sequences (*Figure 1—figure supplement 2*) are marked by vertical lines. Arrows indicate the previously crystallized conserved portion of L1ORF1p (*Khazina et al., 2011*) and the rapidly evolving, non-conserved portion. In contrast to the non-conserved portion, the conserved portion of L1ORF1p can be easily aligned in all mammals (*Boissinot and Sookdeo, 2016*; *Khazina et al., 2011*; *Yang et al., 2014*) (*Figure 1—figure supplement 1B*, *Supplementary file 1*). (**C**) Sequence of the crystallized coiled coil construct (hL1ORF1p-cc), listed as heptads. Heptad positions *a* and *d* are shaded in grey with non-canonical residues in red. Ion-coordinating layers are marked by green circles. In heptad IX, the three-residue insertion corresponding to the stammer is in magenta (this work). Polar residues involved in trimerization motifs and forming stabilizing salt bridges are in blue and circled with interactions shown as connecting lines. Residues in hL1ORF1p-cc that deviate from the human sequence are in brown. See the Materials and methods and *Supplementary file 3* for further construct details. (**D**) Coiled coil propensity and probability of disorder. Coiled coil propensity (blue) was calculated from the alignment of primate sequences using PCoils (*Alva et al., 2016*; *Lupas, 1996*). The probability of disorder (red) was obtained from the human sequence using IUPred (*Dosztányi et al., 2005*).
DOI: https://doi.org/10.7554/eLife.34960.003

The following source data and figure supplements are available for figure 1:

**Source data 1.** Output from Pcoils and IUpred.
DOI: https://doi.org/10.7554/eLife.34960.006

**Figure supplement 1.** Organization of ORF1 proteins in L1 and other non-LTR retrotransposons.
DOI: https://doi.org/10.7554/eLife.34960.004

**Figure supplement 2.** Alignment of closely related primate L1ORF1p sequences.
DOI: https://doi.org/10.7554/eLife.34960.005

Coiled coils are superhelical bundles of α-helices, where each helix is built from repeats of seven amino acids (heptads). In each heptad, the amino acid positions are labeled *a* to *g*, where positions *a* and *d* point towards the center of the bundle and are typically occupied by small, hydrophobic residues. This results in a usually hydrophobic core of the coiled coil with alternating *a*- and *d*-layers. Consequently, the residues in the *a*- and *d*-positions of each heptad are the most critical ones to define and stabilize a coiled coil. Furthermore, charged residues in positions *e* and *g* frequently form stabilizing salt bridges on the surface of the coiled coil (*Lupas et al., 2017*).

In the case of the human L1ORF1p, the presence of a coiled coil forming region was previously identified by sequence analysis (*Hohjoh and Singer, 1996*), but it was difficult to define the boundaries of this coiled coil domain and to align its sequence among mammalian orthologs (*Boissinot and Furano, 2001*; *Boissinot and Sookdeo, 2016*). The identification and crystallization of the RRM domain ultimately revealed that the coiled coil domain extends right to the start of the RRM domain and that mammalian L1ORF1 proteins share seven alignable heptads preceding the RRM domain (*Khazina and Weichenrieder, 2009*). These heptads are numbered I to VII in the C- to N- terminal direction and include two conserved 'R*hxxh*E' trimerization motifs spanning heptads V and VI, where '*h*' designates hydrophobic *a*- and *d*-layers and 'x' stands for any residue (*Kammerer et al., 2005*). Trimerization motifs stabilize the parallel, trimeric structure of a coiled coil through salt bridges that form between glutamates in position *e* and arginines in position *g'* of the preceding heptad. They probably also help to initiate coiled coil formation and to define the correct register for coiled coil assembly (*Ciani et al., 2010*; *Kammerer et al., 2005*). Together with the RRM and CTD domains, heptads I to VII therefore define the alignable or conserved portion of L1ORF1p (*Figure 1A*, *Figure 1B*, *Figure 1C*, *Figure 1—figure supplement 1B*, *Supplementary file 1*). The conserved portion of human L1ORF1p trimerizes as the full length protein and binds and releases nucleic acid substrates, but it is not sufficient to support retrotransposition (*Khazina et al., 2011*).

The remaining, N-terminal portion of L1ORF1p is variable among mammalian orthologs and cannot be consistently aligned (*Figure 1—figure supplement 1B*, *Supplementary file 1*). It is therefore also missing from recently published alignments (*Boissinot and Sookdeo, 2016*; *Yang et al., 2014*). The N-terminal portion of L1ORF1p consists of the presumably disordered NTR, followed by additional heptad repeats that complete the predicted coiled coil domain (*Figure 1D*). It is possible though to unambiguously align the presently active human L1ORF1p sequence with ancestral L1ORF1p sequences reconstructed from the human genome (L1PA1 up to L1PA5) (*Khan et al., 2006*) and with closely related L1ORF1p sequences from the great apes and macaques (*Figure 1—figure supplement 2*). This alignment predicts a coiled coil domain with seven additional heptad repeats (VIII to XIV) and with an insertion of three amino acids in or around heptad IX. Such an insertion disturbs the periodicity of the coiled coil and is called a 'stammer', in comparison to 'stutters', which are insertions of four residues (*Brown et al., 1996*).

Most importantly, the alignment also illustrates the rapid evolution of the N-terminal portion of human L1ORF1p as compared to the rest of the sequence and especially the accumulation of non-conserved residues in the N-terminal half of the coiled coil domain (*Figure 1B*, *Figure 1—figure supplement 2*). This part of the coiled coil domain has previously been claimed to be under positive selection, because it appears to evolve more rapidly than expected from a neutral rate of evolution (*Boissinot and Furano, 2001*; *Khan et al., 2006*). Furthermore, among mammalian L1ORF1ps, the number and regularity of the N-terminal heptads varies considerably (*Figure 1—figure supplement 1B*, *Supplementary file 1*), especially in mice, where heptad duplications and deletions are well documented for the three active L1 lineages (*Sookdeo et al., 2013*) (*Figure 1—figure supplement 1C*, *Supplementary file 1*).

To characterize the molecular properties and functional requirements of the essential but poorly conserved N-terminal portion of L1ORF1p, we therefore took an individual, structure-based approach with the human L1ORF1p.

## The crystal structure of the entire coiled coil domain of human L1 ORF1p reveals malleability of the N-terminal heptads

Sequence analysis suggests the coiled coil domain to begin with residue Y52 of the human L1ORF1p (*Figure 1C*, *Figure 1D*). Considering the high sequence variation even among primate orthologs, it was unclear, however, whether the entire sequence could form one continuous coiled coil, where and how the three amino acid insertion would be accommodated and what would be the structural

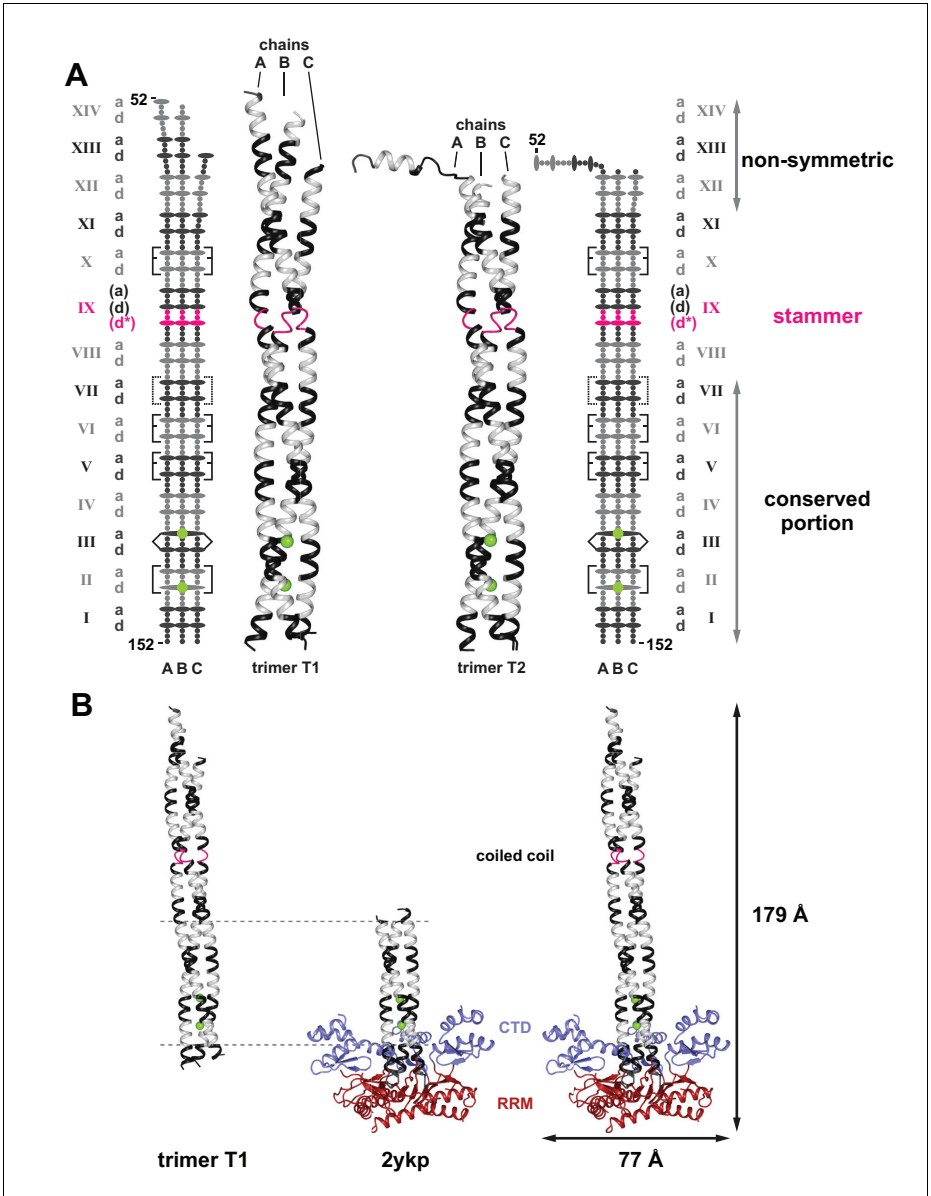

**Figure 2.** Crystal structure of the human L1ORF1p coiled coil domain. (**A**) The two crystallized trimers of the L1ORF1p coiled coil domain. See *Table 1* for data collection and refinement statistics. Structures include the rapidly evolving N-terminal portion of the coiled coil. Structures are shown as three-dimensional ribbons or schematically, with colors as in *Figure 1*. Heptad positions *a* and *d* are symbolized by ovals, the remaining positions as small circles. Rectangular brackets symbolize trimerization motifs or other stabilizing salt bridges (dashed). The pointed bracket symbolizes R135 in position III*d* and its coordination of a chloride ion in the preceding *a*-layer. (**B**) Composite structure model for the human L1ORF1p trimer. Superposition of the coiled coil trimer T1 (residues 52–152) via heptads II-VI (horizontal lines) with a crystal structure of the human L1ORF1p conserved portion (PDB-ID 2ykp, residues 107–223) (*Khazina et al., 2011*) results in a model comprising all parts of human L1ORF1p that are known to be structured. See also *Figure 2—figure supplement 1*.

DOI: https://doi.org/10.7554/eLife.34960.007

The following figure supplement is available for figure 2:

**Figure supplement 1.** Crystallographic details and properties of the coiled coil in the context of the composite model of the L1ORF1p trimer.

DOI: https://doi.org/10.7554/eLife.34960.008

**Table 1.** Data collection and refinement statistics.

| Data collection | |
|---|---|
| Space group | P2$_1$2$_1$2 |
| Cell dimensions | |
| $a$, $b$, $c$ (Å) | 92.2, 250.6, 33.8 |
| α, β, γ (°) | 90, 90, 90 |
| Wavelength (Å) | 1.000 |
| Resolution (Å)* | 125–2.65 (2.72–2.65) |
| $R_{sym}$* | 7.3 (81.3) |
| $I/\sigma I$* | 14.9 (2.2) |
| Completeness (%)* | 99.8 (98.4) |
| Redundancy* | 5.4 (5.5) |
| **Refinement** | |
| Resolution (Å) | 125–2.65 |
| No. of reflections | 23830 |
| $R_{work}/R_{free}$ (%) | 20.8/23.8 |
| No. of atoms | |
| Protein | 4851 |
| Ligand/ion | 4 |
| Water | 64 |
| B-factors (Å$^2$) | |
| Protein | 87.6 |
| Ligand/ion | 69.3 |
| Water | 58.3 |
| R.m.s deviations | |
| Bond lengths (Å) | 0.010 |
| Bond angles (°) | 0.82 |

*Highest resolution shell is shown in parenthesis.
DOI: https://doi.org/10.7554/eLife.34960.009

and functional consequences of the numerous non-canonical residues in the predicted *a*- and *d*-layers.

We therefore tried to obtain a detailed crystal structure and indeed, a bacterially expressed fragment of human L1ORF1p, encompassing the entire coiled coil domain (hL1ORF1p-cc) crystallized with two trimers (T1 and T2) present per asymmetric unit, yielding two slightly differing structures of the trimeric human L1ORF1p coiled coil domain at 2.65 Å resolution (*Figure 2A*, *Figure 2—figure supplement 1A*, *Figure 2—figure supplement 1B*, *Table 1*).

Trimer T1 forms an extended rod with an overall length of 150 Å. For the polypeptide chains A and B, all 14 heptads are found in the electron density in a continuously helical, extended conformation. The variable coiled coil sequences therefore indeed extend the previously characterized C-terminal heptads (I-VI) and retain threefold symmetry up to heptad XI. In heptad XII, chain C begins to deviate and breaks the threefold symmetry, and then becomes untraceable in the electron density in heptad XIII. Chains A and B instead continue a helical packing with likely support from crystal contacts (*Figure 2A*).

Trimer T2 is highly similar to trimer T1 for heptads II to XI (r.m.s.d. for C$_\alpha$ atoms = 0.781 Å), but, in comparison to trimer T1, heptad XII also remains roughly threefold symmetric. Furthermore, heptads XIII and XIV are untraceable in electron density in the case of chains B and C. In the case of chain A, heptad XIII locally unwinds and loses its α-helical structure, whereas heptad XIV is still α-helical but bent by ~90° with respect to the threefold axis, making crystal contacts with a T1 trimer from a neighboring asymmetric unit (*Figure 2A*).

Apparently therefore, the N-terminal heptads of the coiled coil domain are deformable and can switch between an α-helical and an unwound state. In particular, the non-canonical K62 and F69 in the *d*-layers of heptads XIII and XII might be difficult to maintain in a three-fold symmetric state. In solution, these heptads hence might preferably engage in alternating binary interactions between two of the three chains, resulting in a dynamic structure at the N-terminal end of the coiled coil domain rather than in the formation of a stable rod.

## The coiled coil domain of human L1ORF1p is characterized by a sharply localized stammer and flanking trimerization motifs

A thorough analysis of the molecular contacts and a computational analysis of coiled coil parameters reveals a mixture of stabilizing and destabilizing interactions along the sequence of the coiled coil domain (*Figure 3*, *Figure 3—figure supplement 1*). Most strikingly, in heptad IX, there is a sharply localized distortion in the helical geometry of both the coiled coil bundle and of the individual poly-peptide chains. The distortion is caused by the stammer, which can be precisely assigned to residues M91, E92 and L93. These three residues form an extra $3_{10}$-helical turn between positions *d* and *e* of heptad IX and create an additional hydrophobic core layer (*d**) at L93. (*Figure 2A*, *Figure 3A*, *Figure 3D*, *Figure 3—figure supplement 1A*, *Figure 3—figure supplement 1D*). Consequently, the individual helices are locally overwound and stretched in concert with a strong increase in the left-handed supercoiling of the bundle (*Figure 3B*, *Figure 3—figure supplement 1B*). Trimeric stammer structures have previously been discussed only in synthetically designed coiled coil environments (*Hartmann et al., 2016*; *Hartmann et al., 2009*) and occur much less frequently in natural coiled coils than stutters, which, in structural terms, are easier to accommodate (*Lupas et al., 2017*). Also in the coiled coils of mammalian L1ORF1 proteins, stutters occur more often, such as in murine L1ORF1p (*Figure 1—figure supplement 1B*, *Figure 1—figure supplement 1C*, *Supplementary file 1*).

In general, stammers are considered to have an unfavorable, destabilizing effect on the respective coiled coil structure (*Lupas et al., 2017*), consistent with the local increase in the averaged atomic B-factors of the two crystallized trimers of the human L1ORF1p coiled coil (*Figure 3C*, *Figure 3—figure supplement 1C*). Additionally, this coiled coil hosts a series of non-canonical and non-ideal *a*- and *d*-layers, which are also considered to be destabilizing (*Figure 1C*, *Figure 2A*, *Figure 3D*, *Figure 3—figure supplement 1D*). In particular, these are the distorted *d*-layers in heptads XIII (K62) and XII (F69), the cysteine and threonine-containing *d*-layers of heptads XI and X, and the cysteine-containing *a*-layers of heptads VII and VI. Finally, there are the previously described ion-coordinating layers of heptads III and II (*Khazina et al., 2011*). Chloride-binding asparagines (heptad II, N142) are not uncommon in the *d*-layer of parallel, trimeric coiled coils (referred to as asparagines at *d*-, or short, N@*d*- layers [*Hartmann et al., 2009*]) as they help to define both the trimeric state and the correct register of the three chains. Arginines (heptad III, R135) are much more rarely observed at *d*-layers, and especially the combination with a glycine (G132) in the preceding *a*-layer, where the guanidino groups of R135 coordinate a second chloride ion, is unique so far to the human L1ORF1p (*Khazina et al., 2011*) (*Figure 3D*, *Figure 3—figure supplement 1D*).

The destabilizing effects of the stammer and of the non-ideal core layers are balanced, however, by numerous peripheral interactions between pairs of neighboring polypeptide chains and involving polar residues in positions *b*, *e*, and *g* (*Figure 1C*, *Figure 2A*, *Figure 2—figure supplement 1B*, *Figure 3A*, *Figure 3—figure supplement 1A*). Next to the two consecutive trimerization motifs in heptads V and VI, which are conserved in all mammals, there are two additional, non-conserved trimerization motifs in heptads II and X and a peripheral interaction with inverse polarity in heptad VII, that is with an arginine in position *e* and a glutamate in position *g'*. The trimerization motifs differ at position *b*, where various alternative residues contribute to the motif in three of the four cases (S119, D112, T81 in heptads V, VI, X, respectively, *Figure 3D*, *Figure 3—figure supplement 1D*).

As a result, the stammer is flanked by stabilizing motifs both on its C-terminal and on its N-terminal side, and the non-canonical layers in heptads II, VI, VII and X are hedged by peripheral interactions. It is clear as well that this mixture of stabilizing and destabilizing interactions results in the observed distribution of the crystallographic B-factors along the sequence of the coiled coil domain, reflecting a more malleable structure of the coiled coil on its N-terminal side (*Figure 3C*, *Figure 3—figure supplement 1C*). However, given the high sequence variability in the coiled coil region, there

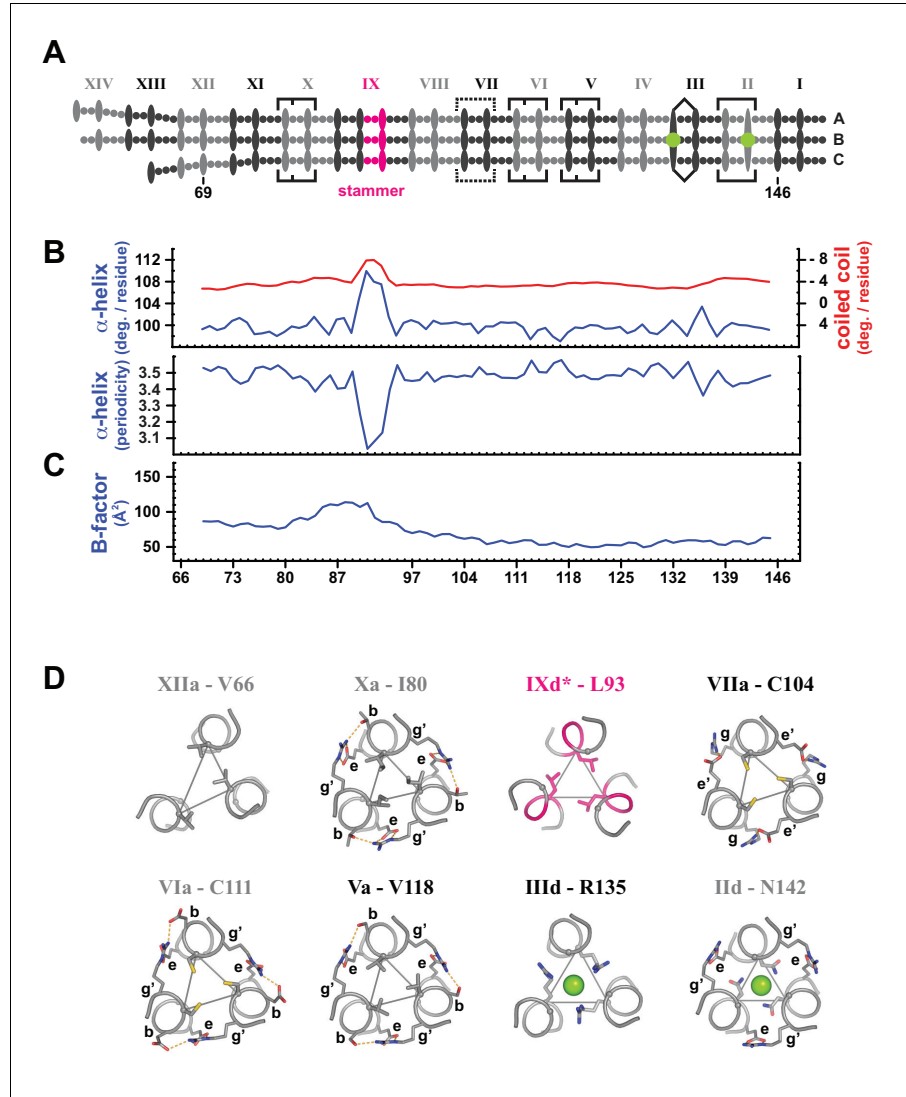

**Figure 3.** Coiled coil parameters and structural details (trimer T1). (**A**) Schematic representation of trimer T1 (see *Figure 2*). (**B**) Coiled coil structural parameters as analyzed in TWISTER (*Strelkov and Burkhard, 2002*) and aligned with the scheme in (**A**). Top panel: Helical geometry of the α-helices (blue) and of the coiled coil bundle (red), expressed in degrees of right-handed rotation per residue. Negative values result from the left-handed geometry of the coiled coil. Bottom panel: Periodicity of the α-helices along the coiled coil axis, expressed in residues per turn. (**C**) Crystallographic B-factor, averaged over the main chain atoms of polypeptide chains A, B and C. (**D**) Individual selected core layers shown as sticks and viewed down the axis of the coiled coil. Heptad positions *a* or *d* of the respective layers are connected by thin lines, and chloride ions are shown as green spheres (heptads II and III). Peripheral heptad positions engaged in trimerization motifs (heptads II, V, VI, X) or other stabilizing salt bridges (heptad VII) are labeled, where *g'* and *e'* mark the positions from the preceding layer. Peripheral stabilization frequently compensates for non-canonical *a*- or *d*-layers. The stammer contributes an additional core layer (IX*d**), geometrically closest to a *d*-type layer (*Strelkov and Burkhard, 2002*). N-terminal heptads are deformable with core layers deviating from three-fold symmetry (XIIa). For trimer T2, see *Figure 3— figure supplement 1*.

DOI: https://doi.org/10.7554/eLife.34960.010

The following source data and figure supplement are available for figure 3:

**Source data 1.** Output from TWISTER and B-factor analysis.
DOI: https://doi.org/10.7554/eLife.34960.012

**Figure supplement 1.** Coiled coil parameters and structural details (trimer T2).
DOI: https://doi.org/10.7554/eLife.34960.011

appear to be many functional combinations of stabilizing and destabilizing interactions, raising the question of how crucial it is to balance the respective effects along the sequence.

## Structural superposition generates a composite model for L1ORF1p including the RRM and CTD domains

The presently determined structures of the coiled coil domain match extremely well with the previously determined structures of the conserved portion of human L1ORF1p (*Khazina et al., 2011*) over heptads II-VI (r.m.s.d. = 0.422 Å over 105 $C_\alpha$ atoms in residues C111-S145, *Figure 2B*). Heptad VII is not completely traceable in the electron density of the conserved portion (*Khazina et al., 2011*) and the C-terminal residues of the coiled coil domain (W150-Y152) are distorted due to crystal packing interactions. Using the overlap for a structural superposition, it is possible to generate a composite model for the human L1ORF1p trimer, which has overall dimensions of 77 Å by 179 Å and comprises the complete coiled coil, RRM and C-terminal domains, that is comprises the conformationally defined region of L1ORF1p (*Figure 2B*).

Coloring the model according to sequence variability illustrates the striking frequency of variable residues in the N-terminal half of the coiled coil, including residues both from core layers and from the surface of the coiled coil (*Figure 2—figure supplement 1C*). Furthermore, the N-terminal half of the coiled coil reveals an alternation of positively charged, neutral and negatively charged surfaces, where an acidic patch at the transition between heptads XI and XII is the most prominent feature. In contrast, the conserved portion of the model is strongly positively charged, especially in the RNA binding clefts between the RRM and C-terminal domains (see also *Khazina et al., 2011*). Notable exceptions are heptads V and VI with their conserved and highly acidic surface (*Figure 2—figure supplement 1D*).

## An NTR peptide is monomeric and disordered and fails to bind the remainder of L1ORF1p

The composite model of the human L1ORF1p trimer (*Figure 2B*) lacks residues M1-N51 and E324-M338, because these residues were missing from the expressed constructs or disordered in the available crystal structures. The C-terminal residues are highly variable or absent in mammalian orthologs (*Figure 1—figure supplement 1B*, *Supplementary file 1*) and can be partially removed (*Alisch et al., 2006*) or also extended with artificial peptide tags without blocking retrotransposition activity (*Goodier et al., 2007*; *Kulpa and Moran, 2005*; *Taylor et al., 2013*). This suggests the C-terminal residues are not functionally required. The 51 N-terminal residues, however, contain functionally relevant phosphorylation sites (*Cook et al., 2015*), but protein constructs including the NTR failed to crystallize.

We therefore used circular dichroism (CD) spectroscopy and analytical size exclusion chromatography to investigate the structure and potential interactions of the NTR (*Figure 4*, *Figure 5*). CD spectroscopy is an excellent method to detect the presence of secondary structure in solution and reveals a purely α-helical spectrum for the hL1ORF1p-cc coiled coil construct (*Figure 4A*). In contrast, a peptide corresponding to the NTR (hL1ORF1p-NTR[H6]) lacks α-helices or β-strands (*Figure 4B*), consistent with the disorder prediction analysis (*Figure 1D*). Furthermore, the coiled coil sequence forms extended trimers in solution as confirmed by multiangle laser light scattering (MALLS, *Figure 4C*), whereas the NTR remains monomeric (*Figure 4D*). Finally, the NTR also fails to interact with the remainder of human L1ORF1p (hL1ORF1p-ΔNTR) when added 'in trans' and tested by size exclusion chromatography (*Figure 4E*, *Figure 4F*, *Figure 4—figure supplement 1*). The NTR also fails to interact with hL1ORF1p-ΔNTR when residues S18 and S27 are substituted by aspartates, mimicking the phosphorylated state of the NTR (*Figure 4—figure supplement 2*). As a result, and in the absence of additional interaction partners, the unstructured NTR peptides appear to be hanging from the deformable and potentially dynamic N-terminal end of the coiled coil domain of the fully assembled trimer, without stable attachment to any of the structured domains.

## The N-terminal heptads of the coiled coil domain are metastable and require the C-terminal heptads for trimerization

The present crystal structures and solution studies show that the coiled coil can form over the entire length of the 14 heptads. However, the seven C-terminal heptads, which are already sufficient for

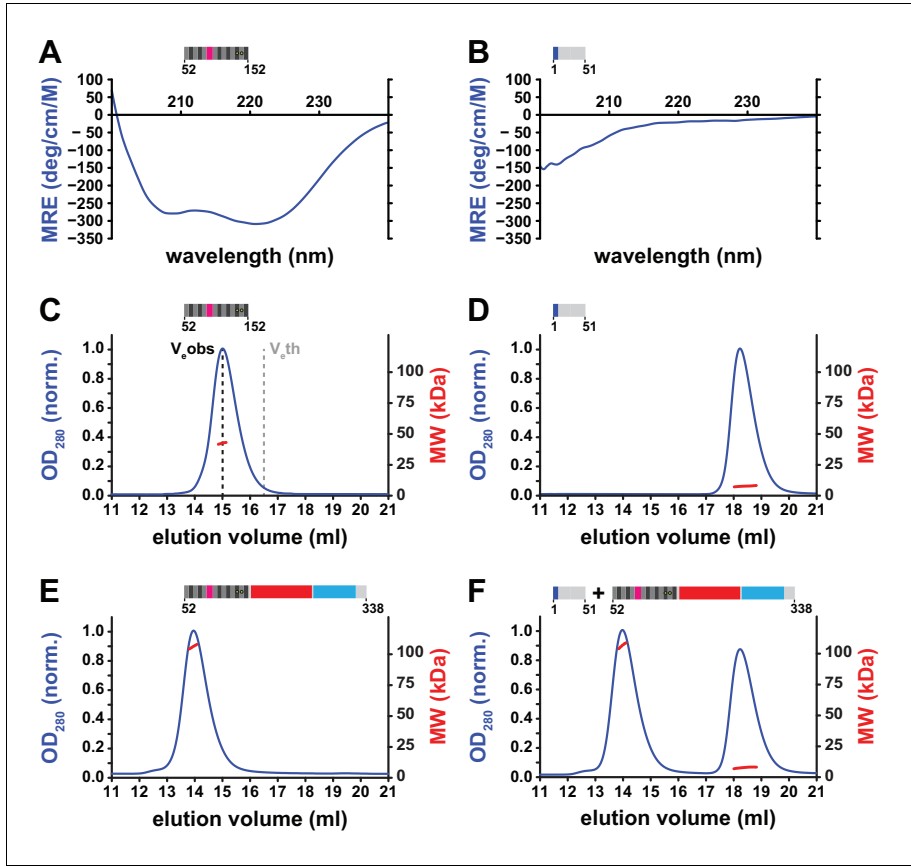

**Figure 4.** Structural properties of the N-terminal region (NTR) of L1ORF1p. (A, B) Circular dichroism (CD) spectroscopy. The spectrum of the L1ORF1p coiled coil domain (A) is typical for an α-helix, whereas the spectrum of the NTR (B) indicates the absence of helices or strands. Ellipticity is calculated per residue as a mean residue ellipticity (MRE), expressed in degrees / (cm x M). (C, D) Size exclusion chromatography followed by multiangle static laser light scattering (MALLS). The molecular weights (MW) determined by MALLS from the elution peaks indicate the coiled coil domain to be trimeric in solution (C, 12.6 kDa per monomer), whereas the NTR remains monomeric (D, 6.7 kDa per monomer). The observed elution volume ($V_e$obs) of the trimeric coiled coil domain is clearly larger than the theoretical elution volume ($V_e$th) of a globular molecule with the same mass. Chromatography was done on a Superdex 200 column and the optical density at 280 nm ($OD_{280}$) was normalized to 1.0 for the maximal absorption observed. (E, F) NTR-binding assay. L1ORF1p is trimeric in the absence of the NTR (E, 34.4 kDa per monomer). The NTR fails to interact with the remainder of L1ORF1p when mixed with the truncated trimer (F). See also *Figure 4—figure supplement 1*, *Figure 4—figure supplement 2*.
DOI: https://doi.org/10.7554/eLife.34960.013

The following source data and figure supplements are available for figure 4:

**Source data 1.** Data from size exclusion chromatography, MALLS and CD spectroscopy.
DOI: https://doi.org/10.7554/eLife.34960.016
**Figure supplement 1.** Gel analysis of the NTR-binding assay.
DOI: https://doi.org/10.7554/eLife.34960.014
**Figure supplement 2.** MALLS and NTR-binding assay with an NTR peptide containing phospho-mimicking residues.
DOI: https://doi.org/10.7554/eLife.34960.015

trimer formation, also are clearly better defined in the electron density map than the seven N-terminal heptads. These show increasingly elevated B-factors and begin to deviate from the threefold symmetry the closer the sequence is located to the amino terminus (*Figure 2A*, *Figure 3C*, *Figure 3—figure supplement 1C*).

We therefore tested whether the variable, N-terminal portion of L1ORF1p would still be able to trimerize in the absence of the conserved portion, but this is clearly not the case. The respective

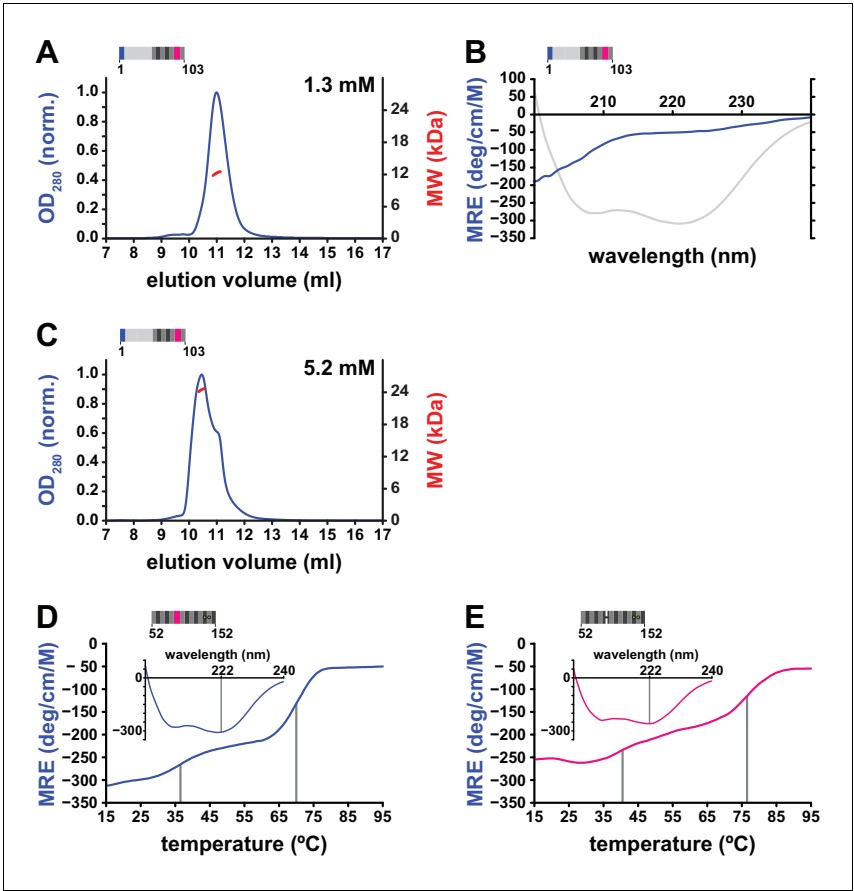

**Figure 5.** Trimerization properties and stability of the coiled coil domain. (**A, B**) Loss of self-association and secondary structure in the non-conserved portion of human L1ORF1p upon deletion of the conserved portion. The non-conserved portion of the human L1ORF1p (12.4 kDa) fails to form oligomers at concentrations up to 1.3 mM, as revealed by size exclusion chromatography and MALLS (**A**). Chromatography was done on a Superdex 75 column and optical density ($OD_{280}$) was normalized to 1.0 for the maximal absorption observed. Furthermore, CD spectroscopy (**B**) shows a total loss of α-helical structure (compare to the spectrum in grey from **Figure 4A**). Ellipticity is calculated per residue as a mean residue ellipticity (MRE), expressed in degrees / (cm x M). (**C**) Dimerization of the non-conserved portion at very high concentrations. At a protein concentration of 5.2 mM, size exclusion chromatography and MALLS reveal a partial dimerization. (**D, E**) Thermal melting curves of the coiled coil domain, monitored by CD spectroscopy at 222 nm. The coiled coil domain comes apart in two distinct steps, one of which occurs at physiological temperature (**D**). The removal of the stammer has a stabilizing effect on the structure and affects both transitions (**E**).

DOI: https://doi.org/10.7554/eLife.34960.017

The following source data is available for figure 5:

**Source data 1.** Data from size exclusion chromatography, MALLS and CD spectroscopy.
DOI: https://doi.org/10.7554/eLife.34960.018

construct (hL1ORF1p-Δcons) remained monomeric at concentrations up to 1.3 mM (**Figure 5A**) and, most surprisingly, is unstructured according to CD spectroscopy (**Figure 5B**). Apparently, folding of the seven N-terminal heptads only occurs when it is triggered by the C-terminal heptads as in the context of the full length L1ORF1p, or possibly, by external binding partners. Consequently, the C-terminal heptads are required for a formation of a continuous coiled coil structure. Alternatively, and at sufficiently high concentration (5.2 mM), the N-terminal portion of L1ORF1p begins to dimerize (**Figure 5C**). Clearly however, the molecular contacts in this dimer must be structurally distinct from the binary interaction of two helices in the trimer, and they could occur in either parallel or anti-parallel orientation.

Given the dependence of homo-trimerization on the C-terminal heptads, we also wondered how the structural stability is affected along the sequence of the coiled coil domain and whether upon thermal denaturation the coiled coil domain would come apart in separate steps or rather cooperatively. Hence, we monitored the loss of α-helical content by CD spectroscopy as a function of increasing temperature and found that indeed, the coiled coil domain (hL1ORF1p-cc) unfolded in a stepwise fashion with two transitions at 37° C and 70° C (*Figure 5D*). In summary, it is therefore reasonable to assume that the first transition reflects the unfolding of exclusively the N-terminal heptads. This leads to a model of the L1ORF1p trimer, where the N-terminal heptads are in a subtle equilibrium between structured and unstructured states and can switch between these states at physiological temperature.

## The presence of the N-terminal heptads and of the stammer are required for L1 retrotransposition activity

To answer the question whether and how much the presence and the biophysical properties of the non-conserved L1ORF1p sequences matter for L1 retrotransposition, we tested a series of L1ORF1p mutants in a well-established, plasmid-based L1 retrotransposition assay in HeLa cells (*Moran et al., 1996*). In this assay, the retrotransposition of a tagged L1 copy into HeLa cell genomic DNA confers an antibiotic resistance. This allows resistant cells to form colonies on a dish, which can then be counted and normalized to wildtype levels (*Figure 6*, *Figure 7*). Expression of the respective L1ORF1p mutants was monitored by western blotting (*Figure 6—figure supplement 1*, *Figure 7—figure supplement 1*).

Because the conserved portion of L1ORF1p was known to be insufficient for activity, we first tested a further extension up to heptad IX, which produces a regular, uninterrupted heptad pattern. However, this construct remained inactive (*Figure 6A*). Next, we reasoned that the NTR with its apparent phosphorylation sites might need to be present, and we generated a series of internal heptad deletions extending over the first seven, five and two of the N-terminal heptads. None of these constructs was active, although at least the latter two were also well expressed (*Figure 6A*). This result is somewhat surprising, because heptad deletions frequently occurred in the evolution of the mammalian L1 element (*Figure 1—figure supplement 1B*, *Figure 1—figure supplement 1C*, *Supplementary file 1*). It suggests that the variable, deformable and non-ideal parts of the coiled coil domain are functionally required in their entirety and consequently, that their ability to alternate between a structured and an unstructured state likely plays a role in the L1 retrotransposition cycle.

In a final step, we therefore exclusively deleted the three stammer residues from heptad IX, generating an uninterrupted, fourteen heptad coiled coil domain with presumably increased stability. This construct too completely failed to retrotranspose (*Figure 6A*). Thermal melting of the respective coiled coil domain construct revealed that, despite the deletion of the stammer (hL1ORF1p-cc$^{\Delta(91-93)}$), the unfolding still was biphasic and hence still not cooperative (*Figure 5E*). However, both unfolding transitions were shifted to higher temperature, indicating that the local deletion of the stammer causes a widespread stabilization over the entire coiled coil domain. Consequently, the human L1ORF1p seems to have evolved to operate in retrotransposition in a rather narrow window of (in)stability.

## L1 retrotransposition depends on non-canonical core layers in the C-terminal heptads and on additional solvent-exposed residues

To further probe the permissive window of coiled coil stability, we tested additional L1ORF1p variants. We primarily targeted unusual core layers, generating single or multiple point mutations at a time (*Figure 6B*, *Figure 6C*).

First, we addressed the ion-containing heptads II and III (*Figure 6B*). Replacement of the unusual R135 at position III*d* with an asparagine (R135N) had the goal to preserve the hydrophilic properties and to support the trimeric state of the coiled coil by allowing for an additional N@*d* layer. This mutation had a rather negligible effect on retrotransposition, whereas the regularizing R135I substitution clearly reduced activity. Also the regularizing G132V substitution in position III*a* detectably reduced retrotransposition, whereas an N142I substitution in position II*d* had a lesser effect. Surprisingly however, the double mutation G132I/R135I and the triple mutation G132I/R135I/N142I (*Khazina et al., 2011*) did not only completely abolish retrotransposition, but also reduced protein

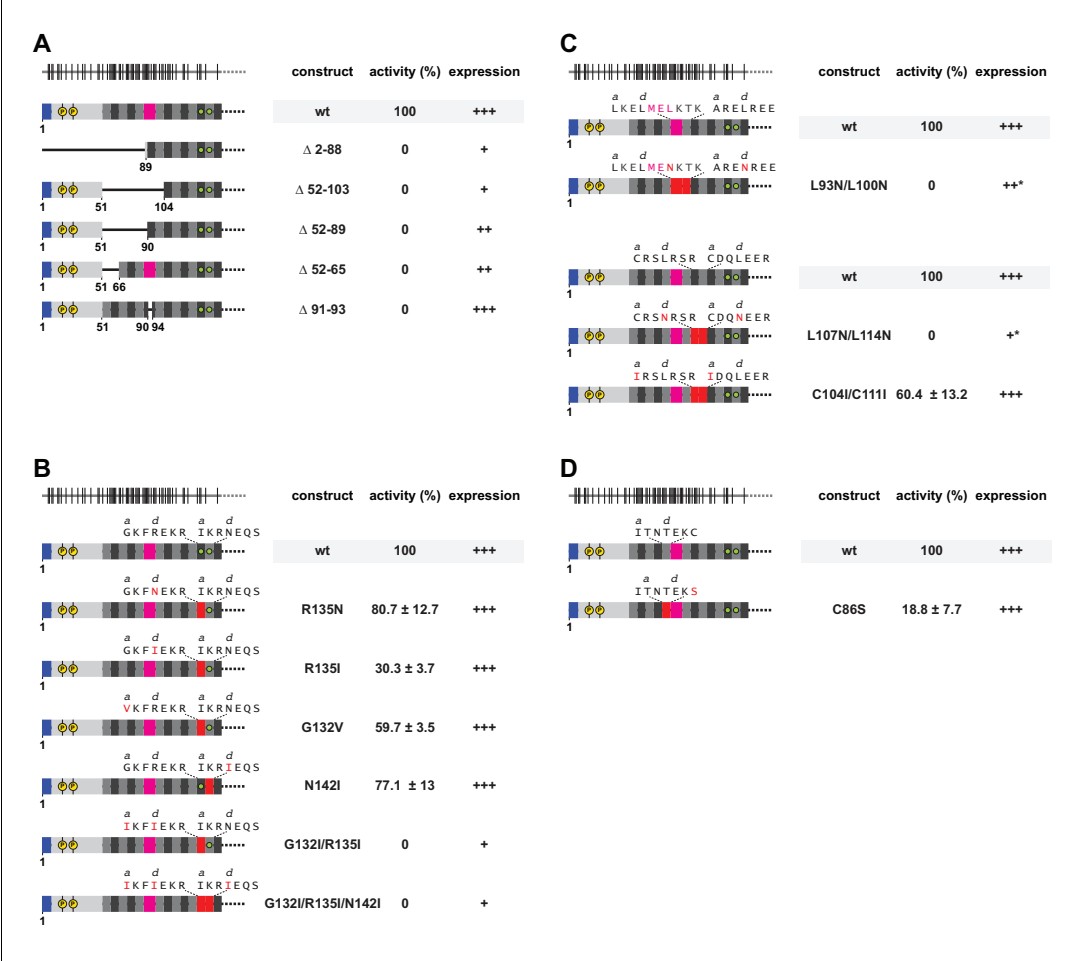

**Figure 6.** Functional importance of a non-ideal coiled coil. (**A**) Deletion analysis of the N-terminal and poorly conserved heptads, monitored by L1 retrotransposition in HeLa cells. Deletions comprise the NTR and/or several complete heptads, or regularize heptad IX by a deletion of the stammer. Retrotransposition activity is calculated with respect to the wildtype (wt) protein, with the mean and standard deviations calculated from three independent replicates. Normal (+++), reduced (++) or poor (+) protein expression levels are marked, as well as an unusual gel migration (*). See also *Figure 6—figure supplement 1*. None of the deletions is tolerated. (**B, C**) Point mutational analysis of core layers. Mutations comprise non-canonical and presumably destabilizing residues in heptads II and III (**B**), as well as canonical and presumably stabilizing leucines in the *d*-layers of heptads VI/VII and VIII/IX (**C**). We also mutated potentially metal-coordinating cysteines (*Ruckthong et al., 2016*) in the *a*-layers of heptads VI/VII (**C**). Mutated amino acids and heptads are highlighted in red. (**D**) Point mutational analysis of a peripheral cysteine in heptad position X*g*.

DOI: https://doi.org/10.7554/eLife.34960.019

The following source data and figure supplement are available for figure 6:

**Source data 1.** Analysis of L1 retrotransposition.
DOI: https://doi.org/10.7554/eLife.34960.021
**Figure supplement 1.** Western blots for the expression analysis of L1ORF1p coiled coil variants.
DOI: https://doi.org/10.7554/eLife.34960.020

levels markedly (*Figure 6—figure supplement 1B*). Presumably, the low protein abundance is caused by a faster degradation of these variants and either due to an improved recognition of the rigidified trimer conformation by the proteolytic machinery or, alternatively, due to an increased mis-assembly of the coiled coil domain in a wrong register.

Second, we addressed heptads VI to IX, which contain a previously investigated series of leucine-containing *d*-layers (*Figure 6C*). These leucines had been tested by various combinations of destabilizing alanine or regularizing valine substitutions, which completely aborted retrotransposition (*Doucet et al., 2010*; *Goodier et al., 2007*). We included the leucine positions in our analysis but used hydrophilic asparagines for substitutions, with the goal to support a trimeric coiled coil but without further stabilization of the structure. However, both an L93N/L100N double mutation in

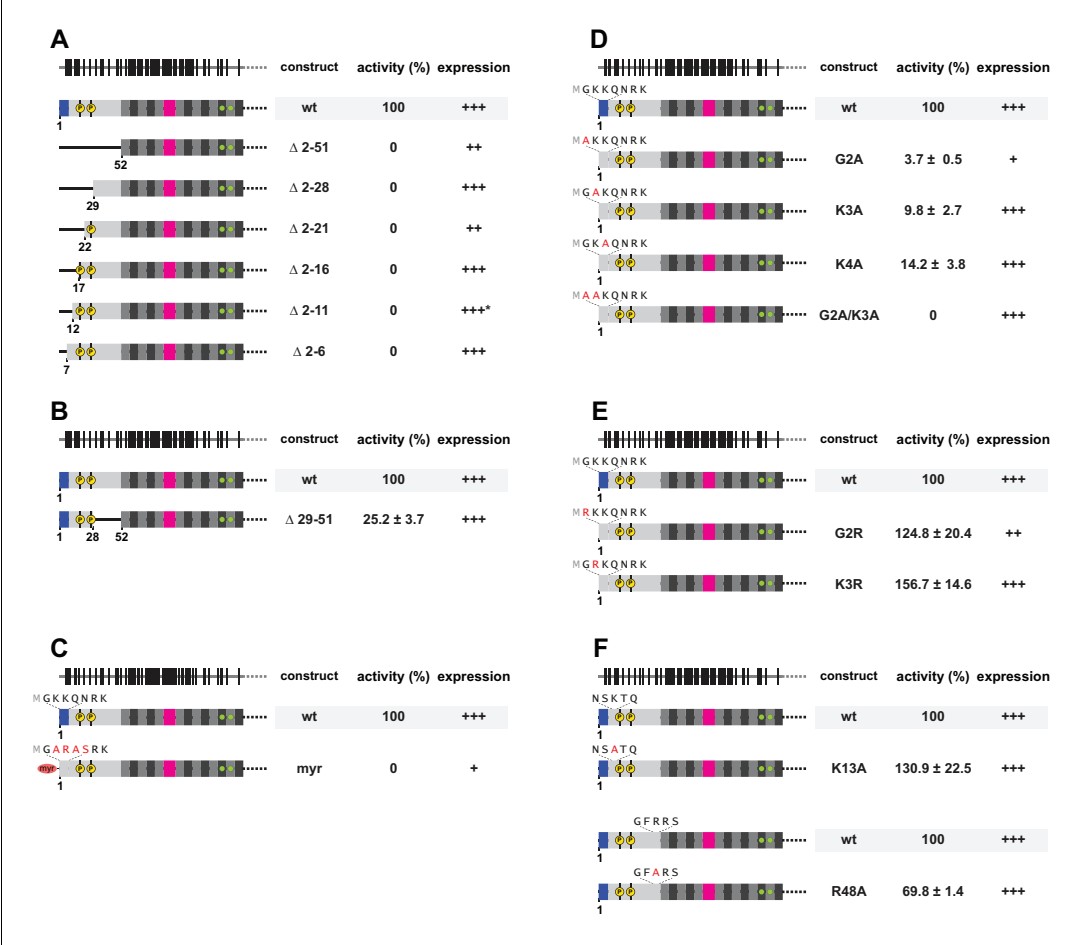

**Figure 7.** Requirement for a positively charged N-terminus on L1ORF1p. (**A**) Systematic deletion analysis of the NTR, monitored by L1 retrotransposition in HeLa cells. Retrotransposition activity is calculated with respect to the wildtype (wt) protein, with the mean and standard deviations calculated from three independent replicates. Normal (+++), reduced (++) or poor (+) protein expression levels are marked. See also **Figure 7—figure supplement 1**. None of the deletions is tolerated. (**B**) Internal deletion in the NTR, permissive for retrotransposition. (**C**) Sequence conversion into an N-terminal myristoylation signal, abolishing retrotransposition. (**D**) Alanine scan of the three N-terminal residues, indicating the importance of these sequence positions. In the wildtype sequence, each of the three N-terminal residues carries a positive charge (after enzymatic removal of the N-terminal methionine). (**E**) Charge-preserving mutations, demonstrating the importance of electrostatics over sequence identity. (**F**) Point mutational analysis of internal NTR residues, demonstrating that internally located lysines or arginines are much less critical for retrotransposition.
DOI: https://doi.org/10.7554/eLife.34960.022

The following source data and figure supplement are available for figure 7:

**Source data 1.** Analysis of L1 retrotransposition.
DOI: https://doi.org/10.7554/eLife.34960.024
**Figure supplement 1.** Western blots for the expression analysis of L1ORF1p NTR variants.
DOI: https://doi.org/10.7554/eLife.34960.023

positions IX*d** and VIII*d* and an L107N/L114N double mutation in positions VII*d* and VI*d* remained inactive. The respective proteins were expressed at reduced levels and moreover migrated slightly abnormally on gels (**Figure 6—figure supplement 1B**). In contrast, a C104I/C111I double mutation in positions VII*a* and VI*a* was active in retrotransposition, although detectably reduced. In eukaryotes, cysteines are not very frequent in the *a*-layers of trimeric coiled coils (**Woolfson and Alber, 1995**). They form tris-thiolate sites, which are predisposed for heavy metal ion binding (**Ruckthong et al., 2016**) and which might have special preferences for the neighboring *d*-layers, such as, for example, the absence of β-branched residues. Indeed, this could possibly explain why the *d*-layer leucines cannot easily be exchanged in the human sequence although they are not even conserved among primates (**Figure 1—figure supplement 2**). More generally, our results

demonstrate that both non-canonical and canonical layers of the coiled coil are very sensitive to mutation and interdependent, and that an idealization and stabilization of the coiled coil structure is rather counterproductive with regard to the ability to retrotranspose.

Third and finally, we also mutated a surface residue in the variable part, C86 in position X*g*, which was converted to a serine (C86S). Surprisingly, even this peripheral single atom substitution of a poorly conserved residue strongly reduced retrotransposition (*Figure 6D*). This suggests that the interdependence of coiled coil residues extends beyond the core layers and to the non-conserved N-terminal half of the coiled coil. The high sequence variability among mammalian L1ORF1ps therefore could well result from an internal coevolution of the residues within the coiled coil domain, where a mutation at one position would trigger a compensatory mutation elsewhere in the coiled coil.

## L1 retrotransposition tolerates variation in the length of the N-terminal region but requires positively charged residues at the amino terminus

To learn whether the NTR is similarly sensitive to mutations, we also started with a deletion analysis (*Figure 7A*). Not surprisingly, a deletion of the entire NTR, including the apparent phosphorylation sites, resulted in an inactive L1 element. However, also consecutive extensions of the L1ORF1p toward the original N-terminus did not rescue L1 retrotransposition activity, indicating that either the overall length of the NTR or the first five amino acids behind the first methionine were crucial for activity. We therefore generated an internal deletion, spanning the second half of the NTR (*Figure 7B*). This construct showed reduced but clearly detectable activity and consequently, the N-terminal amino acids in L1ORF1p are key to L1 retrotransposition activity.

Closer inspection of the N-terminal sequence shows an accumulation of positively charged residues, but also a remote similarity to an N-terminal myristoylation signal. We therefore first altered the sequence to convert it to a strong myristoylation signal, substituting M$^1$GKKQNRK with M$^1$GARASRK (*Bologna et al., 2004*). However, the respective protein construct was only poorly detectable and showed no activity at all, indicating that an N-terminal myristoylation does probably not play a positive role in retrotransposition (*Figure 7C*).

Since the N-terminal methionine is very likely removed from the wildtype sequence (*Frottin et al., 2006*), the remaining GKKQNRK sequence turns into a strongly positively charged patch as long as the main chain amino terminus and the lysine side chains remain non-acetylated and free of any other kind of potential modification. We therefore tested single alanine substitutions in the first three positions behind the methionine (G2A, K3A and K4A) as well as a G2A/K3A double mutation (*Figure 7D*). Strikingly, each of the single point mutations strongly reduced L1 retrotransposition, whereas the G2A/K3A double mutation completely abolished it. Notably, however, the single G2A mutation was accompanied by a very low protein level (*Figure 7—figure supplement 1B*). Since L1ORF1p is a known ubiquitination target, the low protein abundance might be rationalized by a context-induced and K3-dependent ubiquitination and degradation (*MacLennan et al., 2017*). But because the G2A/K3A double mutation was expressed at normal levels, we did not follow up on this possibility here any further. Instead, we generated two alternative single point mutations, G2R and K3R, which were well tolerated and, for K3R, even led to a detectable increase in retrotransposition activity (*Figure 7E*). This result shows that it is not the identity of the N-terminal amino acids or the presence of a specific post-translational modification, but rather the presence and accumulation of the positive charges that matter for retrotransposition in this case.

Finally, it is important to note that the positive charges need to be present near the N-terminus of the NTR, because internal substitutions of positive charges (K13A and R48A) did not have a strong effect (*Figure 7F*) and because the addition of N-terminal tags to the natural amino terminus of L1ORF1p is known to abolish retrotransposition activity (*Goodier et al., 2007*; *Taylor et al., 2013*). Therefore, the positively charged N-terminal end of human L1ORF1p emerges as a previously unknown retrotransposition requirement and appears to be a feature that is conserved all the way through the evolution of the mammalian L1 element (*Figure 1—figure supplement 1B*, *Supplementary file 1*).

## Discussion

### Molecular characterization of L1ORF1p

Non-LTR retrotransposition is still poorly understood on a mechanistic level. In particular, it is unclear what are the molecular properties of the diverse ORF1 proteins and how these properties promote essential steps in the retrotransposition cycle. Moreover, ORF1 proteins do not have cellular or viral homologs from which their mechanics and function could be deduced, requiring an individual analysis.

The present work leads to a deeper mechanistic understanding of the human L1ORF1 protein and especially of its previously only poorly characterized and rapidly evolving N-terminal sequences. There are two key findings. First, L1 retrotransposition apparently requires a long, non-ideal and metastable coiled coil with the ability to switch between structured and partially unstructured states. Second, retrotransposition activity also requires a flexible NTR with a strongly positively charged amino terminus. We therefore speculate that adjacent phosphorylation, conformational changes in the coiled coil domain, and/or the bound L1 RNA could regulate the availability of the amino-terminal residues in a cellular context and hence control crucial steps in the L1 retrotransposition cycle.

Our findings reinforce the picture of the L1ORF1p as a delicate and highly flexible protein with functions that clearly go beyond its previously investigated RNA binding and chaperoning functions (*Figure 8*, *Figure 8—video 1*, *Figure 8—figure supplement 1A*). The conserved portion of the coiled coil domain plays a central role for the assembly of the trimer. It is necessary and sufficient to specify and promote trimerization, and it serves as a scaffold for the oriented but flexible attachment of the RRM and CTD domains, which cooperate in binding single stranded RNA substrates (*Khazina et al., 2011*). Importantly, however, the conserved portion of the coiled coil domain also triggers the assembly of the non-conserved portion, which, together with the positively charged amino terminus on the unstructured NTR, fulfills crucial but hitherto poorly contemplated roles in the retrotransposition cycle. These are outlined in the following.

### Rapid evolution of L1ORF1p

A mechanistic requirement to switch between structured and transiently unstructured states imposes opposing constraints on the amino acid sequence of the coiled coil and can explain the presence of untypical core layers or heptad expansions, and hence the conservation of an irregular rather than a canonical coiled coil structure in the evolution of the L1ORF1p. The need to switch between conformational states can also explain the rapid sequence evolution in the N-terminal half of the coiled coil. A faster-than-neutral amino acid substitution rate can arise when an initial mutation that disturbed the finely calibrated balance of stabilizing and destabilizing interactions gets compensated and fixed by another mutation elsewhere in the sequence of the coiled coil. Such an intrinsic cause for the rapid evolution is also supported by engineered coiled coil chimeras generated from reconstructed ancestral and modern human L1ORF1p proteins (*Naufer et al., 2016*). These chimeras functioned only in one of two possible combinations, whereas the original proteins both are fully functional in the retrotransposition assay. External causes for the rapid evolution may independently exist in the form of a coevolving restriction factor or of an evasive interaction partner from the host (*Daugherty and Malik, 2012*). The fact, however, that highly diverged L1ORF1ps from the mouse or a reconstructed L1ORF1p from the megabat promote human L1 retrotransposition in HeLa cells (*Wagstaff et al., 2011*; *Yang et al., 2014*) argues against the existence of an evasive interaction partner and indicates a remarkable autonomy of L1ORF1p to promote retrotransposition independently of the host cell's molecular environment.

### Parallels to other dynamic coiled coil proteins

Our findings reinforce parallels of the L1ORF1p to other coiled coil proteins, where coiled coil formation also allows for the switch between two (or more) conformational states, including the exposure or capture of functional peptide sequences. Classical examples are viral membrane fusion proteins (*Chen et al., 1999*; *Kobe et al., 1999*; *Weissenhorn et al., 1997*), best studied for the influenza hemagglutinin. Here, refolding and homotypic trimeric coiled coil formation exposes N-terminal and hydrophobic peptides that mediate membrane fusion (*Lin et al., 2014*; *Skehel and Wiley, 2000*). L1ORF1p lacks any such hydrophobic sequences, but the positively charged amino terminus

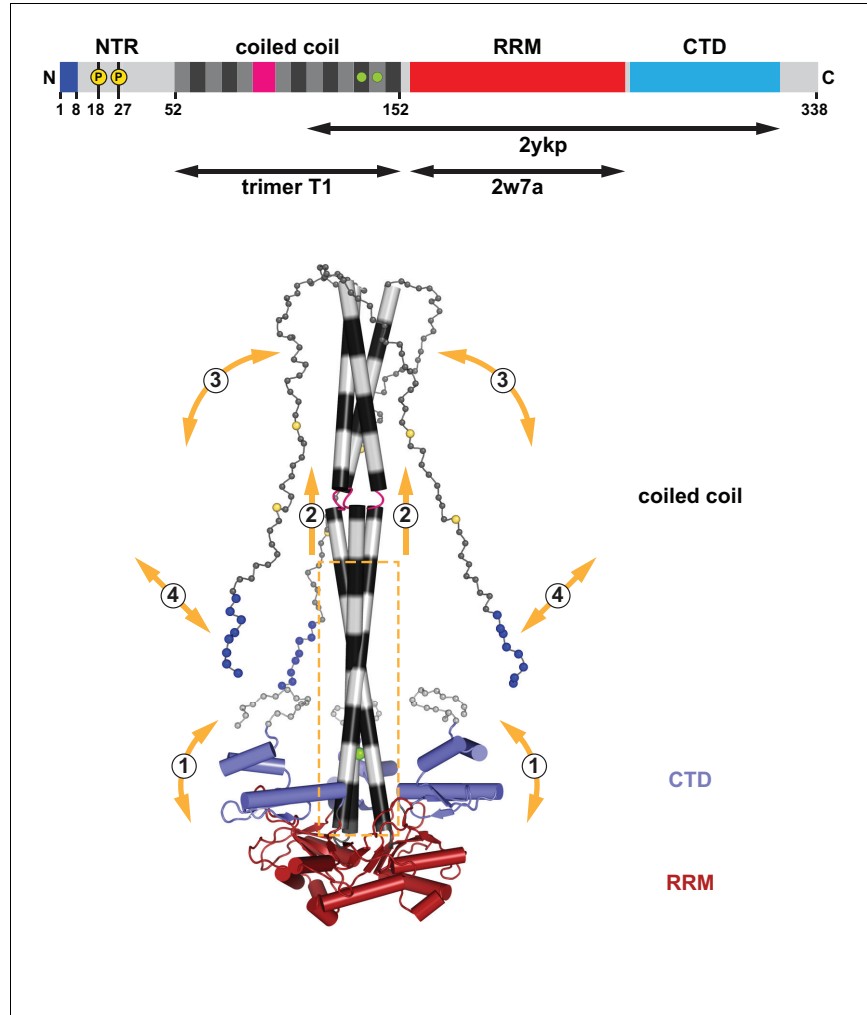

**Figure 8.** Assembly and flexibility of the human L1 ORF1p trimer. A complete structural model for the human L1ORF1p trimer was generated by superimposing known crystal structures (ribbons representation with cylindrical helices) and by geometrically restrained free modeling of the unstructured regions as random coil (balls-and-sticks representation). Starting from the composite model in *Figure 2B* (generated from trimer T1, residues 52–152, and PDB-ID 2ykp, residues 107–223) (*Khazina et al., 2011*), the three RRM domains were replaced by the crystal structure of the isolated RRM domain (PDB-ID 2w7a, residues 157–252) (*Khazina and Weichenrieder, 2009*), where all of the connecting loops are defined. The presumably unstructured N-terminal and C-terminal regions were then added for size comparison, where each sphere represents a $C_\alpha$ atom. The conserved part of the coiled coil domain (boxed) is necessary and sufficient for the trimerization of L1ORF1p and serves as a scaffold for the flexible attachment of the CTDs, which cooperate with the RRM domains in L1RNA binding (*Khazina et al., 2011*) (1). The conserved part of the coiled coil domain is also required for the folding and trimeric assembly of the non-conserved part of the coiled coil (2), which can switch between structured and unstructured states (3). Finally, L1ORF1p functionally requires a strongly positively charged N-terminus (blues spheres) on the unstructured NTR. Interactions of the N-terminal amino acids might be regulated (4) by adjacent phosphorylation (yellow spheres), by the conformation of the coiled coil domain and/or by the bound L1RNA. For a video, see also *Figure 8—video 1*.
DOI: https://doi.org/10.7554/eLife.34960.025

The following video and figure supplement are available for figure 8:

**Figure supplement 1.** Hypothetical model for an oligomerization of L1ORF1p trimers into linear arrays and larger meshworks.
DOI: https://doi.org/10.7554/eLife.34960.026

**Figure 8—video 1.** Video of the complete structural model of the human L1ORF1p rotating around the threefold axis.
DOI: https://doi.org/10.7554/eLife.34960.027

might also serve to target lipid bilayers due to their negative surface charge (*Hoernke et al., 2012*; *Kim et al., 1991*). Other examples are the eukaryotic SNARE proteins (*Sutton et al., 1998*), where heterotypic tetrameric coiled coil formation specifies vesicle targeting to membranes and causes signal-dependent vesicle fusion by zipping up in a stepwise fashion (*Gao et al., 2012*; *Jahn and Fasshauer, 2012*; *Südhof and Rothman, 2009*). A final example is the bacterial protein M1 (*McNamara et al., 2008*), which can form dimeric coiled coils in two alternative registers, but where the transient and destabilized intermediate is functionally important for pathogenicity, allowing the capture of fibrinogen-derived peptide sequences (*Stewart et al., 2016*).

## Coiled coil mediated multimerization of L1ORF1p

The thermal melting experiment with the L1ORF1p-derived coiled coil indicates that its N-terminal half can come apart at a physiological temperature, whereas its C-terminal half remains trimeric. Furthermore, when tested in isolation at a high local concentration, the N-terminal sequence of L1ORF1p dimerizes. These observations raise the possibility that trimers of L1ORF1p directly interact with each other at high concentrations, for example during assembly of L1RNPs. The consequence of dimerizing trimers is not only a linear array on the RNA, but rather a three-dimensional meshwork with probably variable regularity (*Figure 8—figure supplement 1B*). Meshwork formation may explain the cytoplasmic 'aggregates' of L1ORF1p that had been observed early on by ultracentrifugation (*Hohjoh and Singer, 1996*; *Martin, 1991*) and by fluorescence light microscopy (*Goodier et al., 2007*; *Martin and Branciforte, 1993*), and also why L1ORF1p appears to 'polymerize' when artificially assembled on long, single stranded DNA (*Naufer et al., 2016*). Electron micrographs of what presumably are perinuclear clusters of L1ORF1p in mutated mouse spermatocytes show an irregular dotted pattern, where 5–6 dots occasionally form semi-closed circles (*Soper et al., 2008*). Intriguingly, when L1ORF1p trimers are assembled into hexameric rings by simple modeling, one obtains a similar diameter of roughly 150 Å (*Figure 8—figure supplement 1B*). A single L1RNA could theoretically accommodate up to 130 trimers (*Khazina et al., 2011*), but this number appears to be considerably lower in purified L1RNPs, obtained from HEK293T cells under stringent salt conditions (*Taylor et al., 2013*). Functionally, meshwork formation might allow L1ORF1p to sequester L1RNA from the host cell environment and to shield it from processes such as deadenylation and decay (*Wahle and Winkler, 2013*) until the L1RNP finally gains access to nuclear chromatin for the reverse transcription and integration steps of the L1 retrotransposition cycle.

## Possible roles for the positively charged N-terminus

The requirement of a positively charged N-terminal peptide sequence came as an unexpected and novel finding here and merits future investigation. At the current stage, we can only speculate on possible functions, but the presence of an essential peptide at the N-terminus in conjunction with an irregular and dynamic coiled coil reinforces mechanistic parallels to viral membrane fusion proteins, where conformational changes regulate the exposure of their fusion peptides (*Skehel and Wiley, 1998*; *White et al., 2008*).

In the case of the L1ORF1p, the positively charged N-terminus could act as a cellular localization or transport signal and/or to target chromatin, especially since certain insect retrotransposons encode L1ORF1p-like proteins with an N-terminal PHD domain (*Metcalfe and Casane, 2014*). PHD domains are frequently found in chromatin reader proteins (*Musselman and Kutateladze, 2011*; *Sanchez and Zhou, 2011*) and also occur in other non-LTR retrotransposon-encoded ORF1ps of different architectural types (*Kapitonov and Jurka, 2003*; *Khazina and Weichenrieder, 2009*). Another possible function of the positively charged amino terminus might be the modulation of RNA binding on the RRM and CTD domains, in particular when it comes to facilitating binding and/or release of L1RNA in the context of remodeling a larger L1RNP.

Finally, positively charged peptides can also target and perturb negatively charged lipid bilayers (*Hoernke et al., 2012*; *Kim et al., 2002*; *Kim et al., 1991*), further extending the analogies with the viral membrane fusion and the eukaryotic SNARE proteins from a purely mechanistic to a truly functional level. Intriguingly indeed, the perinuclear clusters of L1ORF1p in mouse spermatocytes appear to be surrounded by a double membrane (*Soper et al., 2008*), and Horn *et al.* have recently shown a dependence of L1 retrotransposition on an interaction of L1ORF1p with components of the ALIX/ESCRT membrane budding complex (*Horn et al., 2017*). It is therefore not unreasonable to

speculate that L1ORF1p also functions in membrane-related processes and particularly in overcoming the nuclear barrier in non-dividing cells (*Kubo et al., 2006*; *Macia et al., 2017*), where a classical, nuclear pore-mediated entry of the large L1RNPs is rather difficult to conceptualize.

## Functional redundancy among structurally diverse ORF1ps in non-LTR retrotransposons

L1ORF1ps from the mouse and from the megabat can functionally replace the human L1ORF1p in human cells, despite considerable sequence divergence and despite the extreme mutational sensitivity of the protein (*Wagstaff et al., 2011*; *Yang et al., 2014*). Similarly, human L1 sequences function in non-human cell-lines and transgenic mice and rats (*Kano et al., 2009*; *Moran et al., 1996*; *Morrish et al., 2002*; *Muotri et al., 2005*; *Ostertag et al., 2002*). This suggests that L1ORF1ps act rather autonomously and in a fundamental fashion, which does not require a highly specific adaptation to the host species.

Furthermore, these findings also lead to the intriguing question whether ORF1ps with an entirely different type of architecture, such as found in other non-LTR retrotransposons, could functionally replace the human L1ORF1p as well. Although direct experimental evidence is still missing, our early observations (*Khazina and Weichenrieder, 2009*; *Schneider et al., 2013*) and recent large-scale sequence analyses of non-LTR retrotransposons (*Heitkam et al., 2014*; *Ivancevic et al., 2016*; *Metcalfe and Casane, 2014*) increasingly support the hypothesis of functional redundancy and of a 'reticulate' (*Metcalfe and Casane, 2014*) rather than a tree-like evolution of ORF1ps. This means that RNA packaging, multimerization and membrane-targeting could be functions which are shared among most ORF1ps encoded by non-LTR retrotransposons (*Schneider et al., 2013*).

## Outlook

Together with previously published structures and analyses (*Januszyk et al., 2007*; *Khazina et al., 2011*; *Khazina and Weichenrieder, 2009*), the human L1ORF1p clearly emerges as the currently best understood ORF1p among non-LTR retrotransposons. The combined structural and mechanistic insight, and the large number of functional mutations presented in this study will enable future research to identify, distinguish and analyze novel steps in the L1 retrotransposition cycle. Furthermore, it is becoming increasingly clear that there are multiple lines of defense to protect the human genome from the uncontrolled propagation of the L1 element. These include mechanisms to control L1RNA transcription and post-transcriptional mechanisms aiming at L1RNA (*Goodier, 2016*; *Pizarro and Cristofari, 2016*), but also processes that directly target the L1ORF1p and merit further investigation (*MacLennan et al., 2017*).

On an entirely different note, the present work also leads to a deeper understanding of the fundamental principles underlying the evolution, stability and dynamics of a coiled coil in a physiological context. Coiled coils are among the most intensively studied protein folds (*Hartmann, 2017*), can be characterized and described from first principles (*Crick, 1953*; *Lupas et al., 2017*) and have become a preferred target for protein design (*Woolfson, 2017*). The present L1 retrotransposition assay could therefore serve as one of the most sensitive assay systems for testing coiled coil designs in a cellular environment.

Finally, for conditions such as certain human cancers with elevated L1 retrotransposition (*Burns, 2017*; *Hancks and Kazazian, 2016*; *Scott and Devine, 2017*), it might become feasible and desirable to develop synthetic small molecules or synthetic peptides (*Modis, 2008*), with the goal to target the stability and function of the coiled coil and thereby to prevent further damage to the genomic DNA by L1 insertion.

## Materials and methods

### Sequence analysis

L1 sequences were retrieved, translated and aligned from the following sources. The human L1.3 sequence (*Dombroski et al., 1993*; *Sassaman et al., 1997*) corresponds to the NCBI accession L19088.1. Ancestral human L1 sequences are from Khan et al. (*Khan et al., 2006*) and the currently active mouse L1 lineages (A1, Tf1, Gf1) are from Sookdeo et al. (*Sookdeo et al., 2013*). Mammalian L1 sequences are found in Boissinot et al. (*Boissinot and Sookdeo, 2016*), and the reconstructed

megabat sequence (NCBI KF796623.1) is from Yang et al. (*Yang et al., 2014*). Individual accession numbers for the reconstruction of primate L1ORF1p sequences are listed in *Supplementary file 2*. PCoils (*Lupas, 1996*) as integrated in the MPI Bioinformatics Toolkit (*Alva et al., 2016*) and IUPred (*Dosztányi et al., 2005*) were used for assigning coiled coil propensity and the probability of disorder, respectively.

## Sample preparation

The DNA sequences encoding purified fragments of the human L1ORF1 protein, hL1ORF1p-NTR[H6] (M[1]–N[51]-HHHHHH), hL1ORF1p-Δcons (GPHM[1]–E[103]), hL1ORF1p-cc (GPHMS[53]–Y[152]), hL1ORF1p-cc[Δ(91–93)] (GPHMS[53]–Y[152] lacking residues 91–93) and hL1ORF1p-ΔNTR (GPHMS[53]–M[338]) are derived from the L1.3 sequence (*Dombroski et al., 1993*; *Sassaman et al., 1997*). They were PCR-amplified from a plasmid harboring a M121A/M125I/M128I triple mutation. The residues substituting the three methionines correspond to the respective residues in the murine sequence, do not reduce human L1 retrotransposition activity, but avoid aberrant initiation of bacterial translation (*Khazina et al., 2011*). The sequence encoding hL1ORF1p-NTR[H6] was inserted into the pET15b expression plasmid (Novagen). The sequence encoding hL1ORF1p(DD)-NTR[H6] with phospho-mimicking aspartates (S18D/S27D) was obtained by site-directed mutagenesis. The sequences encoding hL1ORF1p-Δcons, hL1ORF1p-cc, hL1ORF1p-cc[Δ(91–93)] and hL1ORF1p-ΔNTR were inserted into the pnEA-pH expression plasmid, which provides an N-terminal and removable hexa-histidine tag (*Diebold et al., 2011*). All plasmids are listed in *Supplementary file 3*.

Proteins were expressed in the *Escherichia coli* strain Rosetta 2 (DE3) (Novagen) at 20°C overnight. All constructs were purified from cleared cell lysates apart from hL1ORF1p-cc[Δ(91–93)], which was solubilized from inclusion bodies with the addition of 6M guanidinium hydrochloride. After an initial Ni$^{2+}$-ion affinity step, the removable hexa-histidine tags were cleaved overnight with recombinant human rhinovirus 3C (HRV3C) protease, and hL1ORF1p-ΔNTR was further purified by a heparin affinity step. Finally, all constructs were purified by size exclusion chromatography using a Superdex 75 column (GE Healthcare, Chicago, Illinois) for hL1ORF1p-NTR[H6], hL1ORF1p(DD)-NTR[H6], hL1ORF1p-Δcons, hL1ORF1p-cc and hL1ORF1p-cc[Δ(91–93)], and a Superdex 200 column (GE Healthcare) for hL1ORF1p-ΔNTR. Concentrated protein samples were flash-frozen in gel filtration buffer (10 mM HEPES, pH = 7.5, 300 mM NaCl, 1 mM DTT) and stored at −80°C for further use.

## Crystallization

Initial crystals of hL1ORF1p-cc (45 mg/ml in gel filtration buffer) were obtained in several conditions by sitting drop vapor diffusion (18° C) mixing 0.2 µl of protein solution with 0.2 µl of reservoir solution over an 80 µl reservoir. Crystals were optimized by manual screening around several initial conditions and flash frozen in liquid nitrogen with additional cryoprotection.

The best-diffracting crystal (2.65 Å resolution, *Table 1*) was obtained over a reservoir of 0.1 M HEPES (pH = 7.0), 0.15 M (NH$_4$)$_2$SO$_4$ and 12% PEG 2000. It was grown in a sitting drop by mixing 0.5 µl reservoir solution and 0.5 µl protein solution at a concentration of 22 mg/ml, suspended over a reservoir of 66 µl. Cryoprotection was achieved by shortly soaking the crystal in reservoir solution supplemented with glycerol to a final concentration of 20%.

## Data collection and refinement

Diffraction data were collected at 100 K on a Pilatus 6M detector (DECTRIS, Baden-Daettwil, Switzerland) on beamline PXII (X10SA) of the Swiss Light Source (SLS), Villigen, Switzerland. Data were processed and scaled in spacegroup P2$_1$2$_1$2, using XDS and XSCALE (*Kabsch, 2010*). The structure was solved by molecular replacement using PHASER (*McCoy et al., 2007*) from within the CCP4 package (*Winn et al., 2011*) and with a search model containing nine heptads of a trimeric coiled coil. The search model was created by N-terminally extending the known structure of the six C-terminal L1ORF1p heptads (PDB-ID: 2ykp, residues 111–152) (*Khazina et al., 2011*) with an additional three heptads of polyalanine sequence. Two copies of the search model were found in the asymmetric unit. This structure was then improved and extended by iterative cycles of model building in COOT (*Emsley et al., 2010*) and refinement using REFMAC (*Murshudov et al., 2011*) from the CCP4 package. Final refinement rounds were done using BUSTER (*Bricogne et al., 2016*). The diffraction data and refinement statistics are summarized in *Table 1*.

## Crystal structure analysis

The stereochemical properties for the structures were verified with MOLPROBITY (*Chen et al., 2010*), and coiled coil parameters were analyzed using TWISTER (*Strelkov and Burkhard, 2002*). Sequence conservation was mapped to the protein structure using ProtSkin (*Denisov et al., 2004*) and illustrations were prepared in PyMOL (http://www.pymol.org) with the APBS plugin (*Baker et al., 2001*) to visualize the electrostatic surface potential.

## Analytical size exclusion chromatography and MALLS

Analytical size exclusion chromatography coupled to multiangle static laser light scattering (MALLS) was done in gel filtration buffer and essentially as previously described (*Khazina et al., 2011*; *Khazina and Weichenrieder, 2009*). Protein concentrations ranged from 0.3 mM to 5.2 mM in the case of hL1ORF1p-Δcons. Size exclusion chromatography was done on a Superdex 200 (10/300 GL) column, apart from hL1ORF1p-Δcons, which was analyzed on a Superdex 75 (10/300 GL) column. MALLS was done using miniDAWN TREOS and Optilab rEX instruments (Wyatt Technologies, Santa Barbara, California) and the associated software (Astra from Wyatt Technologies) for molecular weight determination.

## Circular dichroism spectroscopy

Circular dichroism (CD) measurements were done at a protein concentration of 0.15 mg/ml in gel filtration buffer without DTT, on a JASCO J-810 spectropolarimeter (JASCO, Easton, Maryland) equipped with a thermoelectric temperature controller. Spectra were recorded using a 0.1 cm path cuvette at a 1 nm band width with response of 2 s. A scanning speed of 100 nm/min and a data pitch of 0.1 nm were used. Thermal denaturation was monitored at 222 nm with a temperature ramp of 1°C/min and a data pitch of 0.5°C. Ellipticity calculation, buffer subtraction and smoothing was done in the software provided by JASCO. The mean residue ellipticity (MRE) was then calculated accounting for protein concentration and sequence length. In *Figures 4* and *5*, the MRE is expressed in units of degrees / (cm x M), where the molar concentration refers to the number of amino acids rather than protein molecules. One degree / (cm x M) equals 100 degrees x $cm^2$/dmol.

## Retrotransposition of L1 variants in HeLa cells

To score the L1 retrotransposition frequency of L1ORF1p mutants, we adapted a well established cell culture assay (*Moran et al., 1996*) that relies on a plasmid-based L1 reporter construct (pJM101/L1.3) (*Moran et al., 1996*; *Sassaman et al., 1997*) and yields G418-resistant HeLa cell colonies only upon a successful retrotransposition. Mutants of the L1 reporter construct were generated by site-directed mutagenesis and are listed in *Supplementary file 3*. DNA sequencing was used to verify that the desired mutations were the only changes in the L1 reporter construct.

Depending on the L1ORF1p variant and its pre-scored activity, HeLa cells were grown and transfected either in standard six-well plates or in 6 cm dishes. Transfection efficiency was monitored with the help of a luciferase reporter vector (pCIneo-Rluc-ΔSV40neo, *Supplementary file 3*) (*Lazzaretti et al., 2009*) that was co-transfected with each L1 construct. Each series of experiments always included the wildtype L1 reporter construct as a reference. Cells were split 48 h after transfection. In the case of six-well plates, one half of the cells was grown for 12–13 days in DMEM containing G418, and the other half of the cells was used to measure luciferase activity levels on day 3 after transfection. In the case of the 6 cm dishes, a third of the cells was seeded into 10 cm dishes for G418 selection, and another third of the cells was seeded into six-well plates for a subsequent luciferase activity measurement. The G418-resistant HeLa cell colonies were fixed and stained with Giemsa, colony numbers were scored, and the retrotransposition frequency was determined as the number of G418-resistant colonies per number of transfected cells. In *Figures 6* and *7*, the L1 retrotransposition activity is calculated with respect to the wildtype reporter plasmid, with the mean and standard deviations calculated from three independently replicated series of experiments. HeLa cells were provided by Elisa Izaurralde and tested for the absence of *Mycoplasma* using a 'MycoAlert' kit (Lonza, Basel, Switzerland).

## Protein expression of L1ORF1p variants in HeLa cells

To monitor protein expression levels of L1ORF1p mutants, HeLa cells were transfected with modified L1 reporter plasmids encoding C-terminal HA-tags on the respective L1ORF1p variants (*Supplementary file 3*). HA-tags were inserted by site-directed mutagenesis and DNA sequencing was used to verify that the HA-tag was the only change in the L1 reporter construct.

HeLa cells were seeded in six-well plates at a density of $0.75 \times 10^6$ cells per well and transfected after 24 h. L1 reporter plasmids were co-transfected with plasmid pT7-EGFP-C1-MBP (*Supplementary file 3*) (*Lazzaretti et al., 2009*) to express a GFP-MPB fusion protein as a transfection control. As a reference, each series of experiments always included the wildtype L1 reporter construct with an HA-tagged L1ORF1p. Empty plasmid (pcDNA3.1) served as a negative control and endogenous tubulin was detected as a gel loading control.

Cells were lysed 48 h post-transfection in a lysis buffer containing 20 mM HEPES (pH = 7.6), 150 mM NaCl and 0.4% Igepal-CA630. The protein concentration in the lysates was quantified using the Bradford reagent (Bio-Rad, Hercules, California). Equivalent amounts of total protein from the lysates were loaded on a polyacrylamide gel for electrophoresis, followed by transfer to a nitrocellulose membrane and probing with antibodies. Monoclonal HRP-conjugated anti-HA antibody (Roche, Basel, Switzerland, RRID:AB_390917, 1:5000) was used to probe for HA-tagged L1ORF1p. Monoclonal anti-GFP antibody (Roche, RRID:AB_390913, 1:2000) and monoclonal anti-tubulin antibody (Sigma Aldrich, St. Louis, Missouri, RRID:AB_477583, 1:5000) were used to probe for GFP-MBP and tubulin, respectively. Polyclonal anti-mouse IgG-HRP (GE Healthcare, RRID:AB_772193, 1:10000) was used as a secondary antibody. Western blots were developed with the ECL Western Blotting Detection System (GE Healthcare) according to the manufacturer's recommendations and protein expression levels were classified as normal (+++, more than 70% of wildtype), reduced (++, between 70% and 30% of wildtype) or poor (+, less than 30% of wildtype).

## Accession numbers

The atomic coordinates and structure factors have been deposited in the Protein Data Bank under accession number 6FIA.

## Acknowledgements

We are grateful to Elisa Izaurralde, Stefan Grüner, Tobias Raisch, Lara Wohlbold and Marcus D Hartmann for comments on the manuscript. We thank Regina Büttner and Gabriele Wagner for excellent technical assistance. We also thank Duygu Kuzuoğlu-Öztürk, Stefanie Jonas and Marcus D Hartmann for experimental advice and the staff of the Swiss Light Source (Villigen, Switzerland) for assistance during data collection. This work was supported by the Max Planck Society.

## Additional information

### Competing interests

Oliver Weichenrieder: Reviewing editor, *eLife*. The other author declares that no competing interests exist.

### Funding

| Funder | Grant reference number | Author |
| --- | --- | --- |
| Max-Planck-Gesellschaft | Open-access funding | Oliver Weichenrieder |

The funders had no role in study design, data collection and interpretation, or the decision to submit the work for publication.

### Author contributions

Elena Khazina, Data curation, Formal analysis, Validation, Investigation, Visualization, Methodology, Writing—original draft, Writing—review and editing; Oliver Weichenrieder, Conceptualization,

Formal analysis, Supervision, Validation, Investigation, Visualization, Methodology, Project administration, Writing—review and editing

### Author ORCIDs
Elena Khazina http://orcid.org/0000-0003-3626-4422
Oliver Weichenrieder http://orcid.org/0000-0001-5818-6248

### Decision letter and Author response
Decision letter https://doi.org/10.7554/eLife.34960.036
Author response https://doi.org/10.7554/eLife.34960.037

## Additional files

### Supplementary files

• Supplementary file 1. Mammalian L1ORF1p sequences used in *Figure 1—figure supplement 1B* Mammalian L1ORF1p sequences are from Boissinot and Sookdeo (*Boissinot and Sookdeo, 2016*) apart from the human sequence (NCBI accession L19088.1) (*Dombroski et al., 1993*), the mouse sequences (*Sookdeo et al., 2013*) and the megabat sequence (NCBI accession KF796623.1) (*Yang et al., 2014*). The conserved and alignable portions of the proteins are in bold letters (*Khazina et al., 2011*), and the coiled coil domains are listed as heptads as in *Figure 1C*. Regular heptads are shaded grey and black and non-heptad interruptions are shaded in magenta, corresponding to the colors in *Figure 1—figure supplement 1B*. Similarly, the NTRs are light grey with the positively charged N-terminal peptides in blue, the RRM domains are in red and the CTDs are in cyan. Coiled coil propensity was calculated using PCoils (*Alva et al., 2016*; *Lupas, 1996*) and used to assign heptads by manual inspection of results from PCoils. Remaining ambiguities in the precise assignment of non-heptad interruptions require finer sequence sampling or experimental validation to be resolved.
DOI: https://doi.org/10.7554/eLife.34960.028

• Supplementary file 2. Individual accessions for primate L1ORF1p. Primate L1ORF1p sequences in *Figure 1—figure supplement 2* are reconstructed consensus sequences with 60% residue identity in the alignments of the listed accessions.
DOI: https://doi.org/10.7554/eLife.34960.029

• Supplementary file 3. Plasmid constructs based on pJM101 (*Moran et al., 1996*; *Sassaman et al., 1997*).
DOI: https://doi.org/10.7554/eLife.34960.030

• Reporting standard 1
DOI: https://doi.org/10.7554/eLife.34960.031

• Transparent reporting form
DOI: https://doi.org/10.7554/eLife.34960.032

### Major datasets

The following dataset was generated:

| Author(s) | Year | Dataset title | Dataset URL | Database, license, and accessibility information |
|---|---|---|---|---|
| Khazina E, Weichenrieder O | 2018 | Structure of the human LINE-1 ORF1p coiled coil domain | https://www.rcsb.org/structure/6FIA | Publicly available at the RCSB Protein Data Bank (accession no. 6FIA) |

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
