## [Decision Letter]

Thank you for submitting your article "Human LINE-1 retrotransposition requires a metastable coiled coil and a positively charged N-terminus in L1ORF1p" for consideration by *eLife*. Your article has been reviewed by three peer reviewers, and the evaluation has been overseen by a Reviewing Editor and Andrea Musacchio as the Senior Editor. One of the reviewers, Jef D Boeke, has agreed to reveal his identity.

The reviewers have discussed the reviews with one another and the Reviewing Editor has drafted this decision to help you prepare a revised submission.

The paper provides new structural information about L1 Orf1, and new indications about function based on the structure and the effects of mutations. Our reviewers were supportive, and in some cases, very supportive, of the paper. There were requests for revisions from all hands, as laid out below.

It will be very important to try to make clear the novel aspects of the findings, especially the advances beyond the earlier work from the same lab. There were also several places where review of the literature was inadequate.

A point by point response to the reviewers' comments will be important.

We look forward to considering a revised draft.

*Reviewer #1:*

This paper presents structures of a portion of the L1 ORF1, an RNA-binding structural protein of the L1 element that assembles RNPs capable of retrotransposition using ORF2 catalytic activities. We are shown a disordered N-terminal region and the coiled-coil. The lab has previously reported the structure of the coiled-coil trimer and the RRM and CTD (Khazina et al., 2011). Indeed, the earlier paper seems to have covered nearly everything reported here; this current story seems to provide a fairly incremental addition to the picture. Much of the presentation consists of reviewing the earlier structures. Here we add the ~50 AA N-terminal sequence (which is disordered) and part of the coiled-coil. There is a notable "stammer" in the coiled-coil. It would be important to lay out as clearly as possible what is new, and what is not new, in the structures.

An extensive set of deletion and substitution mutants were studied to document the importance of the N-terminal region, and other regions, for efficient transposition. While the N-terminus is needed (rather unsurprisingly), the need for its presumed metastability per se is unclear. Basic charged residues are required. What any of the essential sequences of ORF1 are doing in transposition is unclear. The Discussion includes much unwarranted speculation.

*Reviewer #2:*

This is an outstanding manuscript by Khazina et al., which elegantly demonstrates that the amino terminus of human LINE-1 ORF1p forms a metastable coiled coil structure, and that a switch between structured and transiently unstructured ORF1p conformational states is critical for LINE-1 retrotransposition. Importantly, the data also demonstrate that a basic amino acid "patch" at the L1 ORF1p amino terminus is required for efficient retrotransposition, suggest how ORF1p forms larger trimer-trimer "mesh works" that may function in L1 RNP assembly/function, and provide strong evidence that the rapid sequence evolution of the amino terminus of L1 ORF1p results from the need to maintain the "finely calibrated balance of stabilizing and destabilizing interactions" between individual ORF1p molecules (which differs from previous suggestions that the amino terminus of L1 ORF1p evolves rapidly to escape host repressive factors).

In sum, I enthusiastically recommend publication of the study.

*Reviewer #3:*

This is an elegant piece of work combining structure determination and mutagenesis, and bolstered by further biophysical analyses and molecular modeling that unveils novel and fascinating structural information about human ORF1. This culminates in a model for an L1RNP meshwork that has appealing features and may account for the metastable nature of the extended N-terminal coiled coil domain, and its ability to form partially two-strand structures as seen in the crystal. The importance of the molecular composition of the complete coiled coil domain for the full protein's metastability and what that might mean for functionality, L1 activity, and the evolution of other ORF1 proteins is well supported and speculated upon, but not overly so. The major conclusions are well supported and I have no major substantive concerns.

I would really like the authors to present their amino acid sequence alignments for ORF1p in Figure 1—figure supplement 1, in addition to the "cartoon". The authors do it for the primate elements in Figure 1—figure supplement 2, but not for the more distantly related elements. This would be most useful, e.g. for evaluating the conservation of the basic patch at the N-terminus across a wider phylogenetic swath.

---

## [Author Response]

Reviewer #1:

This paper presents structures of a portion of the L1 ORF1, an RNA-binding structural protein of the L1 element that assembles RNPs capable of retrotransposition using ORF2 catalytic activities. We are shown a disordered N-terminal region and the coiled-coil. The lab has previously reported the structure of the coiled-coil trimer and the RRM and CTD (Khazina et al., 2011). Indeed, the earlier paper seems to have covered nearly everything reported here; this current story seems to provide a fairly incremental addition to the picture. Much of the presentation consists of reviewing the earlier structures. Here we add the ~50 AA N-terminal sequence (which is disordered) and part of the coiled-coil. There is a notable "stammer" in the coiled-coil. It would be important to lay out as clearly as possible what is new, and what is not new, in the structures.

Indeed, the presented crystal structures partially overlap with previous structures. The overlap permits an unambiguous superposition with the previously crystallized conserved portion of the L1ORF1p trimer and the new structures add ~3x60 additional amino acids (~25% ) to the final composite model of the L1ORF1p trimer.

The new structures reveal novel and unexpected properties and point to entirely different aspects of L1ORF1p function, distinct from the previously characterized RNA binding and chaperoning functions (Khazina and Weichenrieder (2009), Khazina et al. (2011)). Importantly, the new structures reveal snapshots of the non-conserved half of the coiled coil in distinct conformational states that deviate from canonical coiled coil geometry – and our functional analysis suggests that it is precisely this irregularity and meta-stability of the coiled coil that seems to be under selection.

To clarify the difference/overlap between the previous and the present structures, we added precise residue numbers to the respective figure legends and verified that all statements referring to the previous structures and findings are properly referenced in the text.

An extensive set of deletion and substitution mutants were studied to document the importance of the N-terminal region, and other regions, for efficient transposition. While the N-terminus is needed (rather unsurprisingly), the need for its presumed metastability per se is unclear. Basic charged residues are required. What any of the essential sequences of ORF1 are doing in transposition is unclear. The Discussion includes much unwarranted speculation.

We distinguish between the presumably unstructured N-terminal region of L1ORF1p (NTR, residues 1-51) and the metastable N-terminal half of the coiled coil (residues 52-103) for which we previously lacked high resolution structural information. The requirement for a strongly positively charged N-terminus on the NTR is an unexpected and novel finding. Similarly, also the requirement and metastability of the non-conserved N-terminal half of the coiled coil came as a surprise and especially the high sensitivity to mutation, given the poor conservation of the mutated elements.

Because it remains unclear precisely how these novel properties of L1ORF1p contribute to L1 retrotransposition, we took the opportunity to add an extended, admittedly speculative but hopefully stimulating discussion with the goal to bridge thoughts and concepts from different research fields and to delineate future research directions.

Reviewer #3:

I would really like the authors to present their amino acid sequence alignments for ORF1p in Figure 1—figure supplement 1, in addition to the "cartoon". The authors do it for the primate elements in Figure 1—figure supplement 2, but not for the more distantly related elements. This would be most useful, e.g. for evaluating the conservation of the basic patch at the N-terminus across a wider phylogenetic swath.

Please note that the “cartoon” in Figure 1—figure supplement 1B does not reflect a phylogenetic sequence alignment, but is a simple alignment of individual bar diagrams, centered on the C-terminal end of the coiled coil domain. We chose this representation to illustrate the high variability of the non-conserved portion of L1ORF1p as compared to the conserved portion, as a motivation for the present work to characterize the structure and properties of the L1ORF1p variable sequences.

Unfortunately, the non-conserved portion of L1ORF1p cannot be reliably aligned across mammals (see Bossinot and Sookdeo (2016)), whereas for the conserved portions of L1ORF1p there already are several alignments available in the literature (see for example Khazina et al. (2011), Yang et al. (2014), Bossinot and Sookdeo (2016)).

Instead of providing an unreliable sequence alignment of the non-conserved portion, we therefore added an additional supplementary file to the manuscript (new Supplementary file 1), where we provide the individual amino acid sequences used to create the cartoons in Figure 1—figure supplement 1B and Figure 1—figure supplement 1C, and where we colored the amino acids (and heptad repeats) as in the cartoon. Annotated in this way, it is easily possible to compare the different portions of the L1ORF1p sequence across the mammalian homologs, as requested by the reviewer.